# Dynamic Multimodal Activation Steering for Hallucination Mitigation in Large Vision-Language Models

**Jianghao Yin, Qin Chen**[*]**, Kedi Chen, Jie Zhou**[*]**, Xingjiao Wu, Liang He**
East China Normal University
`jhyin@stu.ecnu.edu.cn, qchen@cs.ecnu.edu.cn, jzhou@cs.ecnu.edu.cn`

## Abstract

Large Vision-Language Models (LVLMs) exhibit outstanding performance on vision-language tasks but struggle with hallucination problems. Through in-depth analysis of LVLM activation patterns, we reveal two key findings: 1) truthfulness and visual perception capabilities predominantly engage different subsets of attention heads within the model architecture; and 2) truthfulness steering vectors vary significantly across different semantic contexts. Based on these observations, we propose Dynamic Multimodal Activation Steering, a training-free approach for hallucination mitigation. Our method constructs a semantic-based truthfulness steering vector database and computes visual perception steering vectors, enabling context-aware interventions during inference by dynamically selecting the most relevant steering vectors based on input semantic similarity and applying them to the most influential attention heads. We conduct comprehensive experiments across multiple models and datasets, demonstrating that our approach significantly enhances model performance, outperforming existing state-of-the-art methods.

## 1 Introduction

Large Vision-Language Models (LVLMs) have demonstrated remarkable performance on visual question answering (VQA), image captioning, and related tasks (Liu et al., 2023; 2024c; Bai et al., 2023; Chen et al., 2024; Dai et al., 2023). However, these models suffer from significant hallucination phenomena (Huang et al., 2025; Chen et al., 2025b), manifested as fabricating non-existent objects or incorrectly describing image content (Liu et al., 2024b; Bai et al., 2024). Such hallucinations limit the applicability of LVLMs in downstream applications including autonomous driving (Cui et al., 2024), robotics (Li et al., 2024b), and other safety-critical domains.

Due to the complex architecture of LVLMs, the causes of multimodal hallucinations are diverse. To address these multimodal hallucination issues, numerous approaches have been proposed (Leng et al., 2024; Huang et al., 2024; An et al., 2025; Yin et al., 2024; Liu et al., 2024a), which can be broadly categorized into two classes: training-based and decoding-based methods. Training-based methods primarily focus on constructing less biased datasets to fine-tune LVLMs, such as LRV (Liu et al., 2024a), or employing reinforcement learning to train LVLMs, as demonstrated by RLHF-V (Yu et al., 2024a). The limitations of these approaches lie in their requirements for carefully curated data and substantial computational resources, as well as the need to retrain models separately for different architectures. Decoding-based methods, on the other hand, modify the decoding strategies of LVLMs, such as VCD (Leng et al., 2024) and ICD (Wang et al., 2024c). While these methods avoid the need for training, they often compromise the quality of the generated content (Yin et al., 2025).

More recently, researchers have begun investigating activation engineering (Zou et al., 2023; Li et al., 2023b; Wang et al., 2024b) as an alternative approach to reduce hallucinations through targeted intervention in model representations. ICT (Chen et al., 2025a) is an image-object cross-level trusted intervention method that mitigates model hallucinations by applying noise to both images and objects, thereby enhancing the model's attention to visual information. However, this approach

---

[*]Corresponding authors.

primarily focuses on visual-level interventions, neglecting the multimodal characteristics of LVLMs. VTI (Liu et al., 2025b) intervenes in the hidden states of both the visual encoder and large language model during inference by pre-computing steering vectors for visual and textual modalities. Nevertheless, this method employs fixed steering vectors regardless of input variation, ignoring potential semantic differences across diverse contexts. The uniformly applied steering vectors fail to account for the nuanced semantic variations that exist across different inputs.

To address these challenges, we propose dynamic multimodal activation steering (DMAS), a training-free approach for hallucination mitigation in LVLMs. Our method focuses on two types of attention heads in LVLMs: truthfulness-related and visual perception-related. For truthfulness heads, we explicitly model how truthfulness steering vectors vary across semantic contexts. We cluster data semantically and create sample pairs with and without hallucinations within each cluster. By contrasting attention activations between factual and hallucination-prone samples, we extract truthfulness steering vectors. These vectors are stored alongside their cluster embeddings in a key-value database. For visual perception, we calculate activation differences between noise-free and noisy image inputs to derive perception steering vectors that enhance visual attention. During inference, we dynamically retrieve the most semantically relevant truthfulness steering vector for the input query and apply both truthfulness and visual perception vectors to the top-K attention heads with the largest activation differences. This dual intervention effectively reduces hallucinations. The main contributions of our paper are:

- We investigate activation differences in LVLMs, revealing that truthfulness and visual perception capabilities predominantly engage different subsets of attention heads, and demonstrate that truthfulness vectors vary significantly across different semantic contexts through visualization, indicating the necessity for dynamic rather than static intervention approaches.

- We propose dynamic multimodal activation steering, a training-free method for hallucination mitigation that constructs a semantic-based truthful steering vector database and visual perception steering vector, enabling context-aware interventions during inference by dynamically selecting appropriate steering vectors based on input semantic similarity.

- We conduct comprehensive experiments on multiple models across discriminative tasks and open-ended generation datasets. The experimental results demonstrate that our method achieves significant improvements: increasing total scores by 94.66 on MME and reducing 20.2% hallucinations on CHAIR, outperforming existing state-of-the-art methods. These results highlight the effectiveness of our approach in hallucination mitigation.

## 2 RELATED WORK

### 2.1 LARGE VISION-LANGUAGE MODELS

Large Vision-Language Models (LVLMs) have recently undergone rapid development, achieving excellent performance in image captioning and VQA tasks (Yin et al., 2023; Jin et al., 2024). They typically consist of a vision encoder, a connection layer, and an LLM. As for the vision encoder, the VIT from CLIP (Radford et al., 2021) is commonly used. For the connection layer, some models use simple MLP layers for alignment, such as LLaVA (Liu et al., 2023; 2024c), Shikra (Chen et al., 2023), PandaGPT (Su et al., 2023), etc.; some models use Q-former for alignment, like BLIP2 (Li et al., 2023a), InstructBLIP (Dai et al., 2023), etc.; while others design special architectures. However, these LVLMs suffer from serious hallucination problems, and effectively eliminating hallucinations remains a popular research topic.

### 2.2 HALLUCINATION MITIGATION FOR LVLMS

Recently, numerous approaches have been proposed to mitigate multimodal hallucinations (Liu et al., 2024b; Bai et al., 2024), addressing this issue across three key stages: training, inference, and post-processing. At the training stage, some research concentrates on constructing better data to train models. For example, LRV (Liu et al., 2024a) constructs a high-quality instruction fine-tuning dataset containing balanced positive and negative samples, while other studies introduce reinforcement learning to the multimodal domain to reduce hallucination, such as RLHF-V (Yu et al., 2024a) and RLAIF-V (Yu et al., 2024b). These methods typically require carefully constructed training data

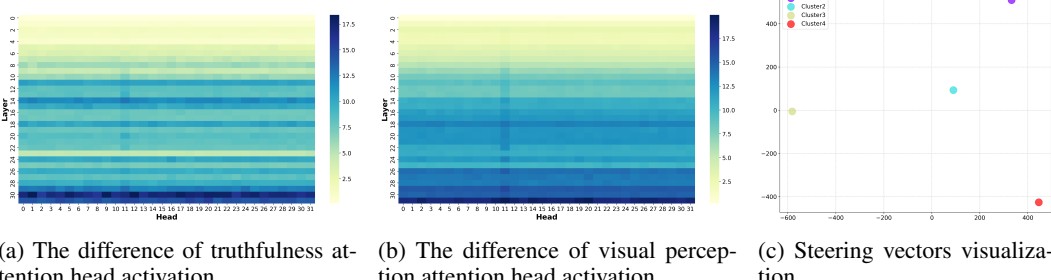

(a) The difference of truthfulness attention head activation.

(b) The difference of visual perception attention head activation.

(c) Steering vectors visualization.

Figure 1: Activation differences in LLaVAv1.5.

and consume substantial computational resources during training. Research on mitigating hallucinations at the inference stage often requires no training. VCD (Leng et al., 2024) uses the distribution from noise-added images and the original output distribution to jointly determine the final distribution to mitigate hallucination. ICD (Wang et al., 2024c) reduces hallucinations by contrasting output distributions between standard and deliberately disturbed instructions. These contrastive decoding methods often compromise the quality of the generated content (Yin et al., 2025). Post-processing approaches correct the generated content from LVLMs to achieve hallucination reduction effects. For instance, LURE (Zhou et al., 2024) constructs a dataset to train a hallucination revisor. However, these methods require the construction of a complex pipeline and increase the time required to obtain final outputs. To overcome these limitations, we propose dynamic multimodal activation steering, a training-free approach to mitigate hallucination in LVLMs by dynamically intervening in attention heads during inference time.

## 3 PRELIMINARY STUDY

To understand the internal mechanisms underlying multimodal hallucinations, we conduct a systematic analysis of attention patterns in LLaVAv1.5 (Liu et al., 2024c) across 3,000 samples from the SEED (Li et al., 2024a) and AMBER (Wang et al., 2023) datasets. Our investigation focuses on identifying which attention heads are most sensitive to truthfulness versus visual perception.

We design two complementary experiments to isolate attention mechanisms responsible for different aspects of multimodal processing. In the first experiment, we examine truthfulness related attention head by contrasting model activations when processing identical visual inputs paired with text prompts either with ground truth or hallucinated answers. This approach enables us to identify attention heads most relevant to truthfulness, we measure how each head's activation changes between truthful and hallucinated content by computing the difference: truthful activation minus hallucinated activation. In the second experiment, we investigate visual perception related attention head by comparing activations between clean images and their noise-corrupted counterparts, calculating activation differences by subtracting the activation values of non-noisy inputs from those with noise. As shown in Figures 1a and 1b, the activation patterns differ significantly between these two experiments. For truthfulness (Figure 1a), the most active attention heads appear predominantly in layer 30. In contrast, for visual perception (Figure 1b), the highest activation differences concentrate in layer 31. These distinct activation patterns provide a foundation for our targeted intervention approach that addresses both aspects simultaneously.

Furthermore, we divide the SEED and AMBER datasets into four semantic clusters and compute the activation differences for each cluster. Using t-SNE to visualize these differences in a two-dimensional space (Figure 1c), we observe a clear separation between clusters, with each occupying a distinct region in the projection space. This separation indicates that truthfulness direction vectors vary significantly across different semantic contexts. The heterogeneity in these patterns suggests that a static intervention approach would be insufficient, as it cannot account for the semantic-dependent nature of hallucinations. This observation directly motivates our dynamic multimodal activation steering method, which can adaptively select appropriate steering vectors based on the semantic content of the input query.

# 4 METHOD

In this section, we introduce dynamic multimodal activation steering. As shown in Figure 2, the method has three steps: the first step is to establish a dynamic truthfulness steering vector database, the second step is to calculate the steering vector for the model's visual perception attention heads, and the third step is to apply dynamic interventions to different attention heads during inference.

## 4.1 TRUTHFULNESS STEERING VECTOR DATABASE

We select the AMBER (Wang et al., 2023) and SEED (Li et al., 2024a) datasets as our data sources and divide the datasets into 4 clusters based on semantics. The questions in these two datasets are in the form of multiple-choice and discriminative questions, making it easy for us to create hallucinated answers for each sample (for discriminative questions, we change the answer to the opposite; for multiple-choice questions, we randomly select an incorrect option). Each cluster $C_i$ comprises the question prompt $T$, visual input $V$, ground truth response $Y_{pos}$, and incorrect response $Y_{neg}$ for every sample.

We input $(V, T + Y_{pos})$ and $(V, T + Y_{neg})$ separately into LVLMs and preserve the attention head activation values of the last token at each layer, denoted as $A_{pos}$ and $A_{neg}$. We define the truthfulness steering vector as the activation difference between non-hallucinated outputs and hallucinated outputs within each cluster according to Equation 1:

$$D_i = \frac{1}{|C_i|} \sum_{j \in C_i} (A_{pos,j} - A_{neg,j}) \tag{1}$$

$|C_i|$ represents the number of samples in cluster $C_i$, and $j$ indexes the samples within the cluster. Subsequently, we apply principal component analysis (PCA) to $D_i$ to reduce insignificant noise, thereby extracting the principal components that influence truthfulness. The magnitude of $D_i$ effectively quantifies the significance of each attention head in governing this specific model behavior.

Next, we construct a truthful steering vector database where the average embedding representation of questions from each cluster serves as the key, with the corresponding steering vector $D_i$ as the value. During inference, our approach dynamically matches the semantic content of the input question to retrieve the most semantically similar steering vector, enabling context-appropriate interventions. We obtain key embeddings in the database and input text embeddings via sentence transformer.

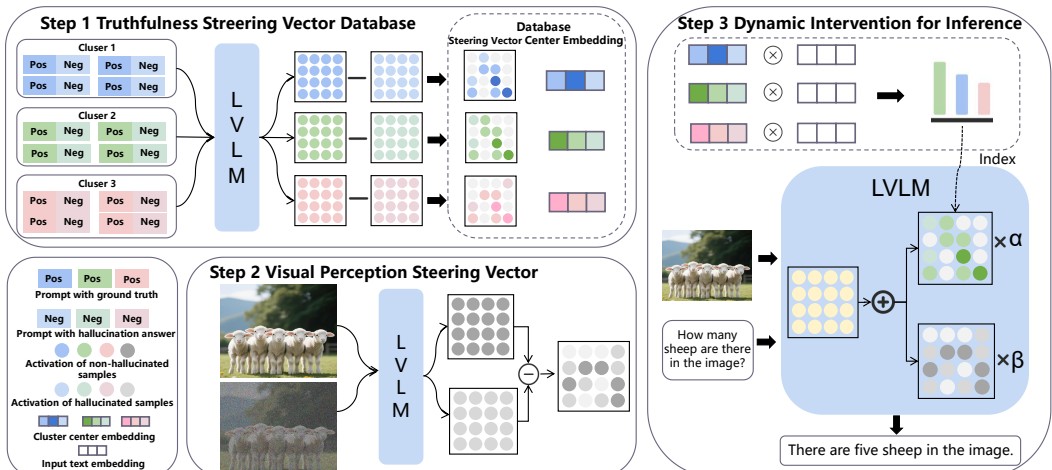

Figure 2: Overview of the DMAS framework.

https://huggingface.co/sentence-transformers/all-mpnet-base-v2

## 4.2 Visual Perception Steering Vector

Given a visual input $V$ and a distorted visual input $V'$ (obtained by adding noise to the image following the forward diffusion process (Ho et al., 2020)), we first input $V$ into an object detector YOLOv11 (Khanam & Hussain, 2024) to obtain objects $O$ present in the image, and insert them into a simple template 'The image depicts {objects}', denoted as $Y_O$. Then we randomly select an equal number of objects $O'$ from a predefined object library within the same object category that are not in $O$, inserting them into template, denoted as $Y_{O'}$. The prompt $T$ is fixed as 'Please describe this image.' Next, we obtain the final inputs $(V, T + Y_O)$ and $(V', T + Y_{O'})$, and input these two samples separately into LVLMs, preserving the attention head activation values of the last token at each layer, denoted as $A_v$ and $A_{v'}$ respectively. We define visual perception steering vector as the activation difference between visual input and distorted visual input according to Equation 2:

$$D_v = A_v - A_{v'} \tag{2}$$

Similarly, we apply PCA to $D_v$ to reduce noise, thereby extracting the principal components most relevant to visual perception.

## 4.3 Dynamic Intervention for Inference

During the inference phase, for a given text input $T$ and visual input $V$, we dynamically retrieve the most appropriate steering vector by computing semantic similarity between the input and each key in database as shown in Equation 3.

$$D_f = D_{\hat{i}}, \text{ where } \hat{i} = \arg\max_i \text{sim}(E(T), Key_i) \tag{3}$$

where $E(T)$ is the embedding representation of the input text, $Key_i$ represents the key embedding for cluster $i$, and $\text{sim}(\cdot, \cdot)$ denotes the cosine similarity function. This process identifies the most relevant truthfulness steering vector for the current input.

To achieve more precise control over model behavior, rather than intervening on all attention heads, we selectively target the most influential heads in both $D_f$ and $D_v$. We define binary mask matrices $M_f$ and $M_v$ as Equation 4:

$$M_{\{f,v\}}^{(l,h)} = \begin{cases} 1, & \text{if } (l,h) \in \text{TopK}(\mathbf{D}_{\{f,v\}}, K) \\ 0, & \text{otherwise} \end{cases} \tag{4}$$

where $(l, h)$ denotes the $h$-th attention head in the $l$-th layer, $\mathbf{D}$ represents the sum of activation differences for each attention head in $D$ and $\text{TopK}(\mathbf{D}_{\{f,v\}}, K)$ returns the indices of the $K$ largest attention heads in either $D_f$ or $D_v$, representing the most influential attention heads for truthfulness and visual perception respectively.

Building upon the standard attention mechanism, we modify the computation for layers where intervention is applied. Our intervention-enhanced computation is formulated as Equation 5:

$$\begin{aligned}
\mathbf{x}^{(l+1)} = \mathbf{x}^{(l)} + \text{Concat}_{(0 \sim H)} \Big[ & \text{Attn}^{(l,h)}(\mathbf{x}^{(l)}) \\
& + \alpha \cdot M_f^{(l,h)} \cdot D_f^{(l,h)} \\
& + \beta \cdot M_v^{(l,h)} \cdot D_v^{(l,h)} \Big] \cdot \mathbf{W}_o^{(l)}
\end{aligned} \tag{5}$$

where $\mathbf{x}^{(l)}$ represents the hidden states at the $l$-th layer, $H$ is the number of attention heads per layer, $\alpha$ and $\beta$ are hyperparameters controlling the intervention strength for truthfulness and visual perception respectively. The binary masks ensure that interventions are only applied to the most influential attention heads, allowing for precise and targeted steering of the model's behavior.

## 5 Experimental Setup

### 5.1 Datasets and Evaluation Metrics

To comprehensively evaluate our proposed approach, we test our method on discriminative tasks, including MME (Fu et al., 2023) and POPE (Li et al., 2023c), as well as on open-ended generation tasks using CHAIR (Rohrbach et al., 2018).

| Model | Method | Existence↑ | Count ↑ | Position↑ | Color↑ | Total Scores↑ |
|-------|--------|-----------|---------|-----------|--------|---------------|
| LLaVAv1.5 | Regular | 175.67 | 124.67 | 114.00 | 151.00 | 565.33 |
| | VCD | 184.66 | 138.33 | 128.67 | 153.00 | 604.66 |
| | OPERA | 180.67 | 133.33 | 123.33 | 155.00 | 592.33 |
| | VAF | **195.00** | 158.33 | 128.33 | 155.00 | 636.67 |
| | DECO | 185.00 | 153.33 | 118.33 | 155.00 | 611.66 |
| | DAMO | **195.00** | 150.00 | **143.33** | 165.00 | 653.33 |
| | AGLA | **195.00** | 153.89 | 129.44 | 161.67 | 640.00 |
| | ICT | 190.00 | **160.43** | 128.67 | 170.00 | 649.10 |
| | Ours | **195.00** | 158.33 | 133.33 | **173.33** | **659.99** |
| QwenVL | Regular | 155.00 | 127.67 | 131.67 | 173.00 | 587.33 |
| | VCD | 156.00 | 131.00 | 128.00 | 181.67 | 596.67 |
| | VAF | 165.00 | **155.00** | **133.33** | 175.00 | 628.33 |
| | ICT | **180.00** | 145.00 | 108.33 | 173.33 | 606.66 |
| | Ours | 170.00 | 145.00 | **133.33** | **185.00** | **633.33** |

Table 1: Results on MME. The best results are shown in bold.

**MME** (Fu et al., 2023) is a comprehensive evaluation benchmark for LVLMs, comprising 14 subtasks. For questions in this dataset, models are required to respond with either 'yes' or 'no'. Following Yin et al. (2024) and Leng et al. (2024), we select 'existence', 'count', 'position', and 'attribute' as the hallucination test sets. Consistent with Fu et al. (2023), we adopt the sum of accuracy and accuracy+ as the evaluation metrics.

**POPE** (Li et al., 2023c) is a benchmark designed specifically to evaluate object hallucination. The benchmark features three sampling strategies of varying difficulty levels: random (randomly sampling nonexistent objects), popular (selecting frequently appearing objects), and adversarial (selecting objects that frequently co-occur with objects present in the image). We report Accuracy, Precision, Recall, F1 Score as the evaluation metrics.

**CHAIR** (Rohrbach et al., 2018) is an open-ended generation task. This benchmark comprises 500 images sourced from MSCOCO (Lin et al., 2014), where LVLMs are required to generate captions for the images, followed by evaluation of hallucinations present in these captions at sentence level $CHAIR_S$ and image level $CHAIR_I$.

## 5.2 BASELINES AND IMPLEMENTATION DETAILS

We validate the effectiveness on mainstream LVLMs: LLaVAv1.5 7B (Liu et al., 2024c) and QwenVL 7B (Bai et al., 2023), and compare DMAS with state-of-the-art methods: ICT (Chen et al., 2025a), AGLA (An et al., 2025), VAF (Yin et al., 2025), VTI (Liu et al., 2025b), DECO (Wang et al., 2024a), DAMO (Wang et al., 2025), VCD (Leng et al., 2024), and OPERA (Huang et al., 2024).

Our method has three key parameters: $\alpha$, $\beta$, and $K$. $\alpha$ and $\beta$ respectively regulate the intensity of interventions for truthfulness and visual perception, while $K$ refers to the intervention on the top $K$ most active attention heads. We set the range of $\alpha$ and $\beta$ to $\{0.5, 1, 2, 3, 4, 5, 6, 7, 8, 9, 10\}$, and the range of $K$ to $\{32, 64, 128, 256, 512, 1024\}$, and employ grid search to determine the parameters. In our experiments, we set the temperature to 0 and top_p to 1. All experiments are conducted on NVIDIA RTX 4090(48GB) GPUs.

## 6 EXPERIMENT

### 6.1 RESULTS ON MME

The results on the MME (Fu et al., 2023) dataset are presented in Table 1. Our method demonstrates significant improvements of 94.66 and 46 points compared to the baseline models LLaVAv1.5 and QwenVL, respectively. On the LLaVAv1.5 model, our approach outperforms the existing state-

of-the-art method ICT (Chen et al., 2025a) by 10.89 points, while on QwenVL, it surpasses the current state-of-the-art method VAF (Yin et al., 2025) by 5 points. Across all subtasks, we observe notable improvements over regular baselines, which can be attributed to our dynamic intervention mechanism that retrieves the most semantically similar steering vector for each query.

## 6.2 RESULTS ON POPE

| Dataset | Setting | Method | Accuracy ↑ | Precision | Recall | F1 Score ↑ |
|---|---|---|---|---|---|---|
| MSCOCO | LLaVAv1.5 | Regular | 81.38 | 88.04 | 72.78 | 79.65 |
| | | VCD | 84.33 | 85.93 | 83.28 | 84.52 |
| | | OPERA | 84.21 | 88.23 | 79.79 | 83.72 |
| | | VAF | 86.90 | 89.43 | 83.77 | 86.47 |
| | | AGLA | 85.82 | 93.78 | 76.83 | 84.44 |
| | | VTI | 86.48 | 90.11 | 82.09 | 85.90 |
| | | ICT | **87.35** | - | - | **87.12** |
| | | Ours | 86.81 | 87.23 | 86.57 | 86.79 |
| | QwenVL | Regular | 83.71 | 93.30 | 72.69 | 81.70 |
| | | VCD | 86.67 | 90.66 | 81.94 | 83.04 |
| | | OPERA | 84.26 | 94.40 | 73.52 | 82.65 |
| | | AGLA | 83.9 | 96.20 | 70.62 | 81.44 |
| | | VTI | 85.18 | 91.31 | 78.18 | 84.08 |
| | | ICT | 87.53 | - | - | 86.98 |
| | | Ours | **87.63** | 87.92 | 87.3 | **87.65** |
| GQA | LLaVAv1.5 | Regular | 78.33 | 79.33 | 79.13 | 79.13 |
| | | VCD | 81.16 | 77.31 | 89.08 | 82.67 |
| | | OPERA | 80.80 | - | - | 83.24 |
| | | VAF | 83.67 | 81.50 | 88.00 | 84.50 |
| | | AGLA | 84.41 | 84.63 | 84.67 | 84.55 |
| | | ICT | **85.27** | - | - | 85.50 |
| | | Ours | **85.27** | 83.86 | 87.51 | **85.63** |
| | QwenVL | Regular | 77.47 | 81.54 | 71.37 | 76.06 |
| | | VCD | 82.48 | 81.73 | 83.93 | 82.77 |
| | | OPERA | 82.74 | - | - | 82.68 |
| | | AGLA | 81.14 | 86.87 | 73.53 | 79.63 |
| | | ICT | 83.28 | - | - | 83.26 |
| | | Ours | **84.40** | 85.19 | 83.53 | **84.32** |

Table 2: Results on POPE. Best results are in bold, and second-best values are underlined.

The experimental results of POPE (Li et al., 2023c) are shown in Table 2. We conduct experiments on MSCOCO (Lin et al., 2014) and GQA (Hudson & Manning, 2019) under random, popular, and adversarial settings. Table 2 presents the average results across these three settings, with detailed experimental results provided in the Appendix. Our method improves LLaVAv1.5's performance on MSCOCO by 5.43% in accuracy and 7.14% in F1 score, while for QwenVL, it achieved improvements of 3.92% in accuracy and 5.95% in F1 score. On GQA, our method enhances LLaVAv1.5 by 6.94% in accuracy and 6.5% in F1 score, and improves QwenVL by 6.93% in accuracy and 8.26% in F1 score. Compared to existing methods, our approach achieves best results in most cases, demonstrating its significant effectiveness in mitigating object hallucination. Notably, while the ICT (Chen et al., 2025a) method applies noise to objects in images to increase the LVLMs' attention to these objects, our method achieves superior performance in most cases without such specialized design elements.

## 6.3 RESULTS ON CHAIR

We evaluate our method on open-ended generation tasks, with experimental results on CHAIR (Rohrbach et al., 2018) presented in Table 3. Our method reduces hallucinations by 20.2 at the

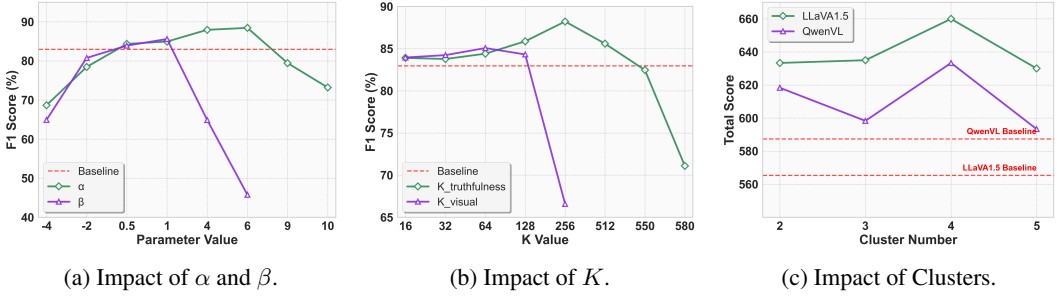

Figure 3: Impact of key hyperparameters.

sentence level (CHAIR$_S$) and by 3.8 at the image level (CHAIR$_I$). Compared to existing methods, our approach reduces sentence-level hallucinations by 5 points over the state-of-the-art method VTI (Liu et al., 2025b), and matching VTI's performance on image-level hallucinations. In summary, our method achieves significant improvements in hallucination mitigation on both discriminative tasks and open-ended generation tasks.

| Method | CHAIR$_S\downarrow$ | CHAIR$_I\downarrow$ |
|--------|--------|--------|
| Regular | 51.0 | 15.2 |
| VCD | 51.0 | 14.9 |
| OPERA | 47.0 | 14.6 |
| DECO | 37.8 | **11.1** |
| AGLA | 43.0 | 14.1 |
| VTI | 35.8 | **11.1** |
| Ours | **30.8** | 11.4 |

Table 3: Results on CHAIR.

| Method | CHAIR | | POPE | |
|--------|-------|-------|------|------|
| | C$_S\downarrow$ | C$_I\downarrow$ | Acc$\uparrow$ | F1$\uparrow$ |
| Ours | **30.8** | **11.4** | **81.70** | **82.47** |
| w/o visual vector | 34.2 | 11.7 | 81.67 | 82.42 |
| w/o truthfulness vector | 42.4 | 13.2 | 81.40 | 82.01 |
| w/o both | 51.0 | 15.2 | 75.08 | 76.06 |

Table 4: Ablation studies on CHAIR and POPE. C$_S$ represents CHAIR$_S$, C$_I$ represents CHAIR$_I$.

## 6.4 FURTHER ANALYSIS

### 6.4.1 ABLATION STUDIES

To demonstrate the effectiveness of using both truthfulness steering vector and visual perception steering vector, we compare the results when utilizing only one intervention at a time. We conduct experiments on LLaVAv1.5 using the CHAIR and POPE. As shown in Table 4, 'w/o visual vector' indicates intervention with only the truthfulness steering vector, while 'w/o truthfulness vector' indicates intervention with only the visual perception steering vector. We observe that even when using only one intervention method, there is a notable improvement compared to the Regular baseline (w/o both). Furthermore, each intervention method exhibits hallucination mitigation effects on both discriminative and generation tasks. The optimal results are achieved when the two interventions are combined.

### 6.4.2 EFFECT OF DYNAMIC INTERVENTION

To validate our designed strategy of dynamically invoking the truthfulness steering vector based on semantics, we compare our method with combining all truthfulness steering vectors into a single fixed steering vector for intervention. We conduct experiments on QwenVL and LLaVAv1.5 using the MME, with results shown in Figure 4. We observe that dynamically invoking steering vectors based on semantics achieves optimal performance across all subtasks. When using a fixed steering vector, the improvement is smaller than with our method, and it even underperforms the original model on the Position subtask of QwenVL, which demonstrates the necessity of our designed dynamic invocation strategy.

### 6.4.3 IMPACT OF HYPERPARAMETERS

In this section, we investigate the impact of key parameters $\alpha$, $\beta$, and $K$ on the experimental results. Here, $\alpha$ and $\beta$ control the intervention strength, while $K$ denotes the number of attention heads receiving intervention. Experiments conducted on QwenVL using the POPE GQA random subset are shown in Figure 3. Figure 3a illustrates the relationship between F1 score and parameters $\alpha$ and $\beta$. When $\alpha$ and $\beta$ are negative, we observe a decrease in F1 score, which effectively represents intervention in the opposite direction, pushing activations toward hallucination. As $\alpha$ and $\beta$ increase, F1 score exhibits an upward trend; however, when $\alpha$ and $\beta$ become excessively large, F1 score shows a

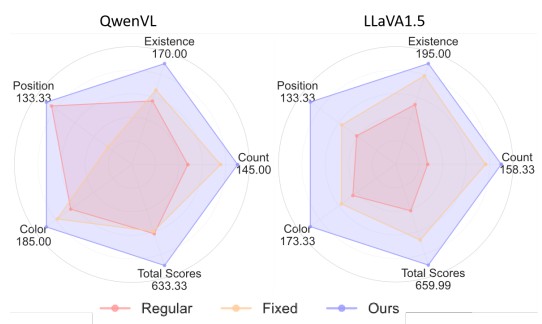

Figure 4: Effect of dynamic intervention.

precipitous decline, indicating that the model's fundamental capabilities become impaired. Figure 3b shows how F1 varies with the number of intervened attention heads, revealing similar patterns for both truthfulness and visual perception attention heads. Few intervened heads produce minimal impact with no significant F1 improvement. As intervention extends to more heads, F1 score increases, but excessive intervention causes a dramatic decline in F1, indicating degradation of model performance.

### 6.4.4 IMPACT OF CLUSTERS

In this section, we investigate the impact of cluster quantity on the performance of our proposed method. We vary the number of clusters across {2, 3, 4, 5} and conduct experiments on both QwenVL and LLaVAv1.5 using the MME benchmark. The experimental results are presented in Figure 3c. We observe that both LLaVAv1.5 and QwenVL achieve optimal performance when the number of clusters is set to 4. When the cluster count is insufficient, the semantic granularity becomes too coarse for effective representation.

### 6.4.5 ANALYSIS OF GENERALITY

To verify the generalizability of our method, we tested our approach on ScienceQA (Lu et al., 2022) which is subject-based VQA dataset and ViQuAE (Lerner et al., 2022) which is a knowledge-based VQA dataset. The accuracy are shown in Table 5. Our method also achieved significant improvements on these datasets. These datasets are completely different from the dataset types we used to construct the steering vector, which demonstrates the generalizability of our method.

| Method | ScienceQA | | ViQuAE | |
|---|---|---|---|---|
| | LLaVAv1.5 | QwenVL | LLaVAv1.5 | QwenVL |
| Regular | 52.75 | 46.41 | 43.38 | 50.09 |
| VTI | 51.46 | - | 42.29 | - |
| ICT | 52.95 | - | 42.47 | - |
| Ours | **62.27** | **48.04** | **56.00** | **54.08** |

Table 5: Generality on ScienceQA and ViQuAE.

## 7 CONCLUSION

This paper proposes dynamic multimodal activation steering, a training-free approach to mitigate hallucination in LVLMs by dynamically intervening in attention head activations. The experiments on multiple benchmarks demonstrate the effectiveness of our method, with LLaVAv1.5 achieving a remarkable 94.66-point improvement on MME and reducing 20.2% hallucinations on CHAIR, outperforming existing SOTA methods. We compare the experimental performance of our proposed semantic-dynamic strategy for steering vector selection against fixed steering vector approaches, demonstrating the effectiveness and necessity of semantic-dynamic steering vector selection.

ACKNOWLEDGMENTS

This research is funded by the National Nature Science Foundation of China (No. 62477010, No.62577022 and No.62307028), the Natural Science Foundation of Shanghai (No. 23ZR1441800), Shanghai Science and Technology Innovation Action Plan (No. 24YF2710100 and No.23YF1426100),Shanghai Qiji Zhifeng Co., Ltd. (2025-GZL-RGZN-01001) and the opening funding of the State Key Laboratory of DisasterReduction in Civil Engineering (Grant No. SLDRCE24-03).

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

# A  APPENDIX

## A.1  AI WRITING ASSISTANCE STATEMENT

Large language models were utilized solely for minor linguistic improvements, including enhanced phrasing and clarity. These tools played no role in content generation, experimental design, data analysis, or interpretation. The authors are entirely responsible for all ideas, results, and conclusions presented in this paper.

## A.2  MORE DETAILS ON CHAIR

In this paper, we report $\text{CHAIR}_S$ and $\text{CHAIR}_I$ as evaluation metrics. The calculation of $\text{CHAIR}_S$ and $\text{CHAIR}_I$ is shown in Equation 6, where we set the maximum number of new tokens to 512 in our experiments.

$$
\begin{aligned}
\text{CHAIR}_S &= \frac{|\{\text{sentences with hallucinated objects}\}|}{|\{\text{all sentences}\}|} \\
\text{CHAIR}_I &= \frac{|\{\text{hallucinated objects}\}|}{|\{\text{all objects mentioned}\}|}
\end{aligned}
\tag{6}
$$

## A.3  RESULTS ON POPE

The complete experimental results on POPE are presented in Table 6. Our method achieves significant improvements across all three experimental settings: random, popular, and adversarial.

| Dataset | Setting | Method | Accuracy ↑ | Precision | Recall | F1 Score ↑ |
|---------|---------|--------|-----------|-----------|--------|-----------|
| MSCOCO | Random | Regular | 83.29 | 92.13 | 72.80 | 81.33 |
| | | VCD | 87.73 | 91.42 | 83.28 | 87.16 |
| | | Ours | **90.03** | 90.51 | 90.03 | **90.02** |
| | Popular | Regular | 81.88 | 88.93 | 72.80 | 80.06 |
| | | VCD | 85.38 | 86.92 | 83.28 | 85.06 |
| | | Ours | **87.33** | 89.16 | 85.00 | **87.03** |
| | Adversarial | Regular | 78.96 | 83.06 | 72.75 | 77.57 |
| | | VCD | 80.88 | 79.45 | 83.29 | 81.33 |
| | | Ours | **83.07** | 82.04 | 84.67 | **83.33** |
| GQA | Random | Regular | 83.73 | 87.16 | 79.12 | 82.95 |
| | | VCD | 86.65 | 84.85 | 89.24 | 86.99 |
| | | Ours | **89.57** | 88.92 | 90.40 | **89.60** |
| | Popular | Regular | 78.17 | 77.64 | 79.12 | 78.37 |
| | | VCD | 80.73 | 76.26 | 89.24 | 82.24 |
| | | Ours | **84.53** | 83.51 | 86.07 | **84.77** |
| | Adversarial | Regular | 75.08 | 73.19 | 79.16 | 76.06 |
| | | VCD | 76.09 | 70.83 | 88.75 | 78.78 |
| | | Ours | **81.70** | 79.15 | 86.07 | **82.47** |

Table 6: Results on LLaVAv1.5. The best results are shown in bold.

## A.4  RESULTS ON AMBER

We conduct an evaluation of LLaVA v1.5 on the AMBER (Wang et al., 2023). AMBER contains both discriminative tasks and generative tasks. The experimental results are shown in Table 7. Our method outperforms existing methods on both discriminative and generative tasks, achieving significant effects in hallucination mitigation.

| Method | Discriminative | | Generative | | |
|---|---|---|---|---|---|
| | Acc↑ | F1↑ | CHAIR↓ | Hal↓ | AMBER SCORE↑ |
| Regular | 67.4 | 71.2 | 11.6 | 47.7 | 79.80 |
| VCD (Leng et al., 2024) | 68.1 | 71.1 | 9.8 | 43.8 | 80.65 |
| ICD (Wang et al., 2024c) | 70.3 | 73.4 | 8.8 | 38.7 | 82.3 |
| IBD (Zhu et al., 2025b) | 69.2 | 72.2 | 9.8 | 42.2 | 81.2 |
| DeFG (Zhang et al., 2025) | 70.2 | 73.0 | 9.1 | 39.9 | 81.95 |
| CICD (Zhao et al., 2025) | 71.1 | 73.1 | 6.6 | 34.8 | 83.25 |
| Ours | **81.9** | **87.2** | **4.9** | **20.9** | **90.01** |

Table 7: Results on AMBER. The best results are shown in bold.

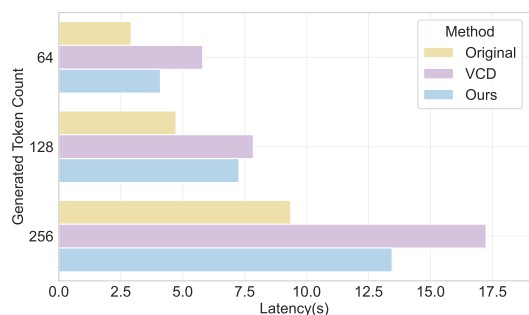

Figure 5: Effect of dynamic intervention.

## A.5  SCALABILITY ANALYSIS ACROSS MODEL SIZES

To verify that our method has hallucination mitigation effects for models of different sizes, we select the discriminative task dataset MME and the generative task dataset CHAIR on LLaVAv1.5 7B and 13B models. The experimental results show in Table 8 that our model achieves significant effects for models of different sizes in both discriminative and generative tasks.

| Model | Method | MME | | | | | CHAIR | |
|---|---|---|---|---|---|---|---|---|
| | | Existence↑ | Count ↑ | Position↑ | Color↑ | Total Scores↑ | $CHAIR_S$↓ | $CHAIR_I$↓ |
| LLaVAv1.5 7B | Regular | 175.67 | 124.67 | 114.00 | 151.00 | 565.33 | 51.0 | 15.2 |
| | Ours | **195.00** | **158.33** | **133.33** | **173.33** | **659.99** | **30.8** | **11.4** |
| LLaVA1.5 13B | Regular | **185.00** | 131.67 | 95.00 | 175.00 | 586.67 | 45.0 | 11.8 |
| | Ours | **185.00** | **158.33** | **103.33** | **180.00** | **626.66** | **38.0** | **10.8** |

Table 8: Scalability Analysis Across Model Sizes on MME and CHAIR. The best results are shown in bold.

## A.6  ANALYSIS OF INFERENCE SPEED

In this section, we investigate DMAS's inference speed. We use LLaVAv1.5 7B and set the generation content lengths to {64, 128, 256} respectively, then compare the inference speed of our method with the original model and VCD. The experimental results show in Figure 5 that our method has faster inference speed compared to the decoding method VCD. VCD's inference latency is almost twice that of the original model, but our model achieves better hallucination mitigation effects while adding only a small amount of inference time.

## A.7  CASE STUDY

To intuitively demonstrate the hallucination mitigation effectiveness of our method, we conduct case studies on LLaVAv1.5. We utilize cases from MME and CHAIR datasets, with results shown

in Figure 6. Our method effectively mitigates multimodal hallucination issues across both VQA tasks and image captioning tasks. For VQA tasks, we present various question types, demonstrating our method's effectiveness in reducing hallucinations at different levels including object, attribute, relation, and count. For image captioning tasks, our method not only generates fewer hallucinations but also maintains the quality of the output content.

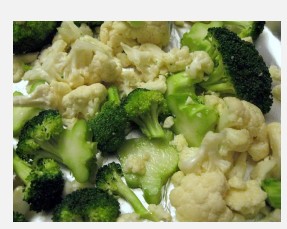

**Question:** Are there a white dish in the image? Please answer yes or no.

**Original model:**
No.

**Ours:**
Yes.

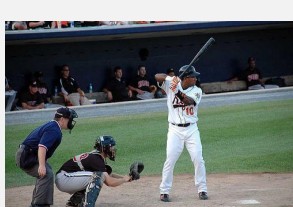

**Question:** Is the cricket bat under the batter's body Please answer yes or no.

**Original model:**
Yes.

**Ours:**
No.

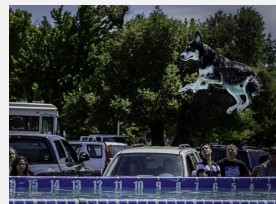

**Question:** Is there only one dog in the image? Please answer yes or no.

**Original model:**
No.

**Ours:**
Yes.

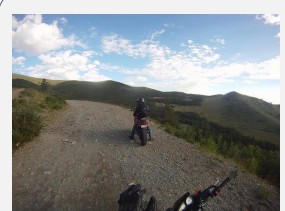

**Question:** Is there a backpack in this image? Please answer yes or no.

**Original model:**
No.

**Ours:**
Yes.

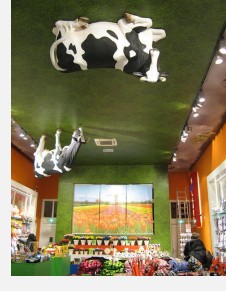

**Original model:**
The image features a large room with a unique design, resembling a barn or a store. The room is filled with various cow statues, including a large cow statue in the center and several smaller cow statues scattered throughout the space. The cow statues are positioned in different areas of the room, creating a visually interesting scene.

**Ours:**
The image features a large, open room with a fake ceiling that resembles a green field. In this room, there are two fake cows hanging from the ceiling, creating a unique and eye-catching display. The room also has a few potted plants placed around the space, adding a touch of nature to the environment.

Figure 6: Case study on LLaVAv1.5.

## A.8 EXPERIMENTS ON MORE BENCHMARKS

To further substantiate the effectiveness of our approach, we conduct extensive experiments across multiple challenging benchmarks, including MMVP (Tong et al., 2024), IllusionVQA (Shahgir et al., 2024), HallusionBench (multimodal subset) (Guan et al., 2024), and MM-Vet (Yu et al., 2024c), evaluating on both LLaVAv1.5 and QwenVL. The results summarized in the Table 9 confirm that our method delivers strong and robust performance across these diverse and complex evaluations.

| Model | Method | MMVP | MM-Vet | IllusionVQA | HallusionBench |
|---|---|---|---|---|---|
| LLaVAv1.5 | Vanilla | 30.3 | 38.9 | 32.6 | 50.2 |
| | ICT | 31.0 | 33.4 | 33.1 | 52.7 |
| | Ours | **54.3** | **39.3** | **35.2** | **53.7** |
| QwenVL | Vanilla | 41.0 | 32.9 | 31.7 | 42.4 |
| | Ours | **43.3** | **34.4** | **33.6** | **42.9** |

Table 9: Experiments on more benchmarks.

### A.9 COMPREHENSIVE ANALYSIS OF THE CONSTRUCTED DATASET

In this subsection, we will explore how the dataset scale and the dataset composition for constructing steering vectors respectively affect the effectiveness of DMAS.

#### A.9.1 DATASET SCALE

To investigate the impact of dataset size on steering vector construction, we systematically reduce the dataset to 75%, 50%, and 25% of its original size and evaluated performance on MME (Fu et al., 2023), AMBER (Wang et al., 2023), and MMVP (Tong et al., 2024). The results are presented in Table 10. The results show that dataset size has minimal impact on performance, with only minor fluctuations across benchmarks. This demonstrates that our method is robust to variations in dataset scale and does not require large amounts of labeled data.

| Scale | MME | | | | MMVP | AMBER | |
|---|---|---|---|---|---|---|---|
| | Existence↑ | Count ↑ | Position↑ | Color↑ | Accuracy↑ | Relation↑ | F1 ↑ |
| 100% | 195.00 | 158.33 | 133.33 | 173.33 | 54.3 | 69.0 | 87.2 |
| 75% | 195.00 | 158.33 | 133.33 | 168.33 | 52.0 | 69.0 | 87.2 |
| 50% | 195.00 | 158.33 | 133.33 | 168.33 | 52.0 | 69.0 | 87.2 |
| 25% | 195.00 | 158.33 | 133.33 | 168.33 | 53.0 | 69.0 | 87.1 |

Table 10: Analysis of dataset scale for steering vectors construction.

#### A.9.2 DATASET COMPOSITION

To explore how different datasets affect steering vector construction, we build databases using two alternative datasets: POPE (Li et al., 2023c) and PHD (Liu et al., 2025a). POPE contains only object existence questions (homogeneous), while PHD includes diverse question types similar to our original datasets. Both databases are constructed using the same scale as our main experiments, and we evaluate performance on MME (Fu et al., 2023), AMBER (Wang et al., 2023), and MMVP (Tong et al., 2024). The results are presented in Table 11. Results show that the POPE-based database underperforms on several MME subtasks and AMBER's relation subtask, which is expected given POPE's limited diversity cannot fully exploit our dynamic approach. In contrast, the PHD-based database achieves comparable results to our original method across all benchmarks. These findings demonstrate two important insights: (1) when using datasets with good question diversity, our method achieves consistent performance regardless of the specific dataset chosen, and (2) this validates the necessity of our dynamic retrieval mechanism for selecting appropriate steering vectors based on semantic context.

| Scale | MME | | | | MMVP | AMBER | |
|---|---|---|---|---|---|---|---|
| | Existence↑ | Count ↑ | Position↑ | Color↑ | Accuracy↑ | Relation↑ | F1 ↑ |
| Ours | 195.00 | 158.33 | 133.33 | 173.33 | 54.3 | 69.0 | 87.2 |
| POPE | 195.00 | 158.33 | 123.33 | 153.33 | 52.3 | 65.1 | 86.0 |
| PHD | 195.00 | 158.33 | 133.33 | 173.33 | 53.7 | 68.3 | 86.9 |

Table 11: Analysis of dataset composition for steering vectors construction.

### A.10 EXPERIMENTS ON DIFFERENT DECODING STRATEGIES

To explore the effectiveness of our method under different decoding strategies, including beam search (beam=2), nucleus sampling (temperature=0.5, topp=0.7) and greedy search, the results on MME and MMVP benchmarks are presented in the Table 12. Our method consistently demonstrates effective hallucination mitigation across all decoding settings, confirming the robustness of our approach.

| Decoding Strategies | Method | MME | | | | | MMVP |
|---|---|---|---|---|---|---|---|
| | | Existence↑ | Count ↑ | Position↑ | Color↑ | Total Scores↑ | Accuracy↑ |
| Beam | Vanilla | 190.00 | 153.33 | 113.33 | 148.33 | 604.99 | 29.7 |
| | Ours | **191.67** | **156.66** | **135.00** | **171.66** | **654.99** | **55.1** |
| Nucleus | Vanilla | 190.00 | 147.78 | 123.89 | 150.55 | 612.22 | 32.3 |
| | Ours | **190.00** | **150.00** | **130.00** | **159.44** | **629.44** | **53.1** |
| Greedy | Vanilla | 190.00 | 153.33 | 113.33 | 148.33 | 604.99 | 30.3 |
| | Ours | **195.00** | **158.33** | **133.33** | **173.33** | **659.99** | **54.3** |

Table 12: Experiments on different decoding strategies.

## A.11 EXPERIMENTS ON MORE MODELS

In this subsection, we test the effectiveness of our method on several advanced LVLMs. We select Qwen2.5-VL (Bai et al., 2025), InternVL3 (Zhu et al., 2025a), Mantis (Jiang et al.), and Idefics3 (Laurençon et al., 2024), and conduct experiments on MME and CHAIR using greedy search. The experimental results are shown in Table 13. Our method remains effective for these advanced LVLMs with different architectures, demonstrating hallucination mitigation effects on both discriminative tasks and open-ended generation tasks, which confirms the scalability of our approach.

| Model | Method | MME | CHAIR | |
|---|---|---|---|---|
| | | Total Scores↑ | $CHAIR_S\downarrow$ | $CHAIR_I\downarrow$ |
| Qwen2.5-VL | Vanilla | 698.33 | 47.4 | 10 |
| | Ours | **718.33** | **43.2** | **9.5** |
| InternVL3 | Vanilla | 698.33 | 28.4 | 7.6 |
| | Ours | **713.33** | **26.4** | **6.9** |
| Mantis | Vanilla | 653.33 | 44.6 | 12.7 |
| | Ours | **668.33** | **34.8** | **9.7** |
| Idefics3 | Vanilla | 665.00 | 49.9 | 8.8 |
| | Ours | **678.33** | **45.2** | **8.2** |

Table 13: Experiments on more models.

## A.12 ABLATION STUDIES ON HALLUCINATION OBJECTS SELECTION STRATEGIES

For visual perception attention heads, we compare two hallucination objects selection strategies: (1) random selection across categories (e.g., people → panda), and (2) hard negatives within the same category (e.g., runner → swimmer). As shown in the Table 14, experiments on MME and CHAIR benchmarks demonstrate that both strategies achieve comparable performance with no significant difference. This indicates that our visual steering vector approach is robust to the choice of negative selection strategy.

| Strategy | MME | | | | | CHAIR | |
|---|---|---|---|---|---|---|---|
| | Existence↑ | Count ↑ | Position↑ | Color↑ | Total Scores↑ | $CHAIR_S\downarrow$ | $CHAIR_I\downarrow$ |
| Random | 195.00 | 158.33 | 133.33 | 173.33 | 659.99 | 31.2 | 10.9 |
| Hard Negative | 195.00 | 158.33 | 133.33 | 173.33 | 659.99 | 30.8 | 11.4 |

Table 14: Ablation studies on hallucination objects selection strategies.

