# OpenReview forum: "Dynamic Multimodal Activation Steering for Hallucination Mitigation in Large Vision-Language Models"
_ICLR.cc/2026/Conference — ICLR 2026 Poster_

### Official Review · Reviewer_KcVa · 2025-10-20

**Soundness:** 3
**Presentation:** 3
**Contribution:** 3
**Rating:** 6
**Confidence:** 3

**Summary:**

This paper aims to address the hallucination problem in Large Vision-Language Models (LVLMs). Through an analysis of internal activation patterns, the authors report two key findings: (1) the attention heads responsible for truthfulness and visual perception are largely disjoint within the model; and (2) the truthfulness steering vectors that guide factual output vary significantly across different semantic contexts.
Building on these insights, the paper proposes Dynamic Multimodal Activation Steering (DMAS) — a training-free, plug-and-play inference-time intervention technique. The method consists of three main stages:
1.	Constructing a Truthfulness Steering Vector Database: Semantic clustering is applied to the data, and within each cluster, activation differences between truthful and hallucinatory samples are computed. These are stored in a database where semantic embeddings serve as keys and truthfulness steering vectors as values.
2.	Computing the Visual Perception Steering Vector: Activation differences between the model’s responses to clean and noise-distorted images are used to derive a steering vector that enhances visual attention.
3.	Dynamic Intervention: During inference, the method retrieves the most semantically relevant truthfulness steering vector from the database based on the input text and combines it with the visual perception steering vector. The combined signal is then applied selectively to the most influential attention heads, effectively reducing hallucinations.
Experimental results demonstrate that DMAS achieves significant improvements over existing approaches across multiple benchmarks and model architectures.

**Strengths:**

Clear problem orientation: The paper provides an in-depth analysis of hallucination issues in LVLMs, particularly revealing the functional tendencies of different attention heads and the semantic dependence of steering vectors through preliminary studies, which offers strong motivation for the subsequent method design.

Broad experimental coverage: The paper conducts comprehensive experiments across multiple models (LLaVA v1.5, QwenVL) and multiple benchmarks (MME, POPE, CHAIR), including detailed ablation studies and case analyses, demonstrating the effectiveness of the method.

Significant performance improvement: Experimental results show that DMAS achieves state-of-the-art performance across multiple tasks, with particularly substantial gains on MME and CHAIR metrics.

**Weaknesses:**

Reliance on synthetic counterfactuals in database construction: The method constructs “hallucinated” answers by flipping or randomly selecting incorrect options, which is relatively easy for multiple-choice or discriminative datasets. However, this approach may not reflect realistic hallucination types in open-ended generation tasks. As a result, the learned “steering vectors” may be biased toward these synthetic patterns.

Dependence on specific labeled data: As noted, the main limitation of this method is that its so-called “training-free” property relies on a pre-constructed steering vector database that depends on labeled data (true/false answer pairs). This severely limits the generality and scalability of the approach. The authors should more transparently discuss this prerequisite and its associated limitations.

High hyperparameter sensitivity: The method introduces several key hyperparameters, including intervention strengths $\alpha$ and $\beta$, the number of intervention heads $K$, and the number of clusters. Analysis in Figure 3 shows that model performance is highly sensitive to these parameters, and inappropriate settings can lead to a sharp drop in performance. This makes hyperparameter tuning costly in practical applications.

**Questions:**

Counterfactual generation: For open-ended datasets (e.g., CHAIR, AMBER), how are hallucinated answers generated? Are they human-labeled, randomly perturbed, or model-generated? How sensitive are the steering vectors to this generation method?

Automatic hyperparameter selection: Is there a lightweight, label-free method to choose (α, β, K) or the number of clusters, to avoid expensive grid search for each new model or domain?

---

> ### Comment · Reviewer_KcVa · 2025-11-25
>
> There is no response for my questions. So I reduce my score.

---

> > ### Comment · Reviewer_KcVa · 2025-11-26
> >
> > Thank you for the response. Most of my concerns have been addressed, so I raise my score and confidence.

---

> ### Author Response · Authors · 2025-11-25
> **Response to Reviewer KcVa: Part 1**
>
> We sincerely thank the reviewer for their constructive comments and insightful suggestions, which have been invaluable in improving our manuscript. We are also grateful for your recognition of the strengths of our work. The revisions have been incorporated in the revised manuscript marked in blue. Please kindly find point-to-point responses below.
>
> > ### **About W1: Reliance on synthetic counterfactuals in database construction:**
>
> Thank you for raising this concern. While our synthetic counterfactuals are constructed from discriminative and multi-choice tasks, Tables 3 and 7 demonstrate that our method effectively mitigates hallucinations in open-ended generative benchmarks as well.
>
> This transferability can be attributed to the underlying mechanism of our approach. The Steering Vector represents the direction from hallucinated to non-hallucinated activations in the model's representation space. By steering activations along this direction, we guide the model toward producing more faithful outputs regardless of task format. Thus, although derived from discriminative and multi-choice tasks tasks, the learned Steering Vectors capture generalizable hallucination patterns that transfer effectively to open-ended generation.
>
> > ### **About W2: Generality, Scalability, and Prerequisites:**
>
> Thank you for raising this important concern. We address it from three complementary perspectives: generality, scalability, and prerequisites.
>
> - **Generality**:
>   - **Evidence in the paper**: To validate the generality of our method, we evaluated it on two datasets not specifically designed for hallucination: ScienceQA and ViQuAE. As shown in Table 5, our method achieves significant improvements on both datasets, demonstrating its effectiveness beyond hallucination-specific benchmarks.
>   - **Additional evidence**: To further address your concern, we conducted additional experiments on four diverse benchmarks: MMVP [1], MMVet[2], IllusionVQA [3], and HallusionBench (multimodal subset) [4] using LLaVA1.5 and QwenVL. The results are presented in the table below. These results across diverse benchmarks further confirm the generality of our approach. This experiment has been incorporated into the Appendix (A.8) of the revised manuscript.
>
> |        | MMVP  | IllusionVQA  | MMVet | HallusionBench |
> |:------:|:-----:|:------------:|:-----:|:--------------:|
> | llava  | 30.33 | 32.64        | 38.90 | 50.16          |
> | VTI    | 30.00 | 30.57        | 29.90 | 46.48          |
> | ICT    | 31.00 | 33.10        | 33.40 | 52.67          |
> | Ours   | **54.33** | **35.17** | **39.3** | **53.73**   |
> | QwenVL | 41.00 | 31.72        | 32.9  | 42.38          |
> | Ours   | **43.33** | **33.56** | **34.40** | **42.90**   |
>
> - **Scalability**
>
>   We demonstrate scalability from two critical dimensions: model size and model architecture.
>
>   - **Scalability across model sizes:** As shown in Appendix A.5 (Table 8), DMAS achieves substantial improvements on both LLaVA1.5-7B and LLaVA1.5-13B across discriminative tasks (MME) and generative tasks (CHAIR), demonstrating consistent effectiveness regardless of model scale.
>   - **Scalability across model architectures:** While the main paper evaluates LLaVA1.5 and QwenVL, we further validated our method on recent state-of-the-art models with diverse architectures: Qwen2.5-VL [5], InternVL3 [6], Mantis [7], and Idefics3 [8]. Results are shown below,  these results demonstrate that DMAS can scale effectively across different architectural designs, confirming its broad applicability. This experiment has been incorporated into the Appendix (A.11) of the revised manuscript.
>
> |            |         | MME                 | CHAIR             |                   |
> |:----------:|:-------:|:-------------------:|:-----------------:|:-----------------:|
> | **Model**  | **Method** | **Total Scores↑** | **CHAIR_S↓**     | **CHAIR_I↓**     |
> | Qwen2.5-VL | Vanilla | 698.33              | 47.4              | 10.0              |
> | Qwen2.5-VL | Ours    | **718.33**          | **43.2**          | **9.5**           |
> | InternVL3  | Vanilla | 698.33              | 28.4              | 7.6               |
> | InternVL3  | Ours    | **713.33**          | **26.4**          | **6.9**           |
> | Mantis     | Vanilla | 653.33              | 44.6              | 12.7              |
> | Mantis     | Ours    | **668.33**          | **34.8**          | **9.7**           |
> | Idefics3   | Vanilla | 665.00              | 49.9              | 8.8               |
> | Idefics3   | Ours    | **678.33**          | **45.2**          | **8.2**           |

---

> ### Author Response · Authors · 2025-11-25
> **Response to Reviewer KcVa: Part 2**
>
> - **Prerequisites**
>
>   We examine the prerequisites from two angles: dataset scale and dataset composition. These experiments has been incorporated into the Appendix (A.9) of the revised manuscript.
>
>   - **Dataset scale:** To investigate the impact of dataset size on steering vector construction, we systematically reduced the dataset to 75%, 50%, and 25% of its original size and evaluated performance on MME, AMBER, and MMVP [1]. The results are presented below. The results show that dataset size has minimal impact on performance, with only minor fluctuations across benchmarks. This demonstrates that our method is robust to variations in dataset scale and does not require large amounts of labeled data.
>
> |       |          MME           |        |          |        |      MMVP      |      AMBER      |          |
> |:-----:|:-----------------------:|:------:|:--------:|:------:|:--------------:|:---------------:|:--------:|
> | Scale | Existence↑ | Count↑ | Position↑ | Color↑ | Accuracy↑ | Relation↑ | F1↑ |
> | 100%  |   195.00   | 158.33 |  133.33   | 173.33 |      54.3      |      69.0       |   87.2   |
> | 75%   |   195.00   | 158.33 |  133.33   | 168.33 |      52.0      |      69.0       |   87.2   |
> | 50%   |   195.00   | 158.33 |  133.33   | 168.33 |      52.0      |      69.0       |   87.2   |
> | 25%   |   195.00   | 158.33 |  133.33   | 168.33 |      53.0      |      69.0       |   87.1   |
>
>   - **Dataset composition:** To examine how dataset choice affects steering vector construction, we built databases using two alternative datasets: POPE [9] and PHD [10]. POPE contains only object existence questions (homogeneous), while PHD includes diverse question types similar to our original datasets. Both databases were constructed at the same scale as our main experiments and evaluated on MME, AMBER, and MMVP. Results show that the POPE-based database underperforms on several MME subtasks and AMBER's relation subtask, which is expected given POPE's limited diversity. In contrast, the PHD-based database achieves comparable results to our original method across all benchmarks. These findings reveal two key insights: (1) datasets with sufficient question diversity enable consistent performance regardless of the specific dataset chosen, and (2) even homogeneous datasets like POPE improve over baselines.
>
> |       |           MME           |        |          |        |      MMVP      |      AMBER      |          |
> |:-----:|:-----------------------:|:------:|:--------:|:------:|:--------------:|:---------------:|:--------:|
> | Scale | Existence↑ | Count↑ | Position↑ | Color↑ | Accuracy↑ | Relation↑ | F1↑ |
> | Ours  |   195.00   | 158.33 |  133.33   | 173.33 |      54.3      |      69.0       |   87.2   |
> | POPE  |   195.00   | 158.33 |  123.33   | 153.33 |      52.3      |      65.1       |   86.0   |
> | PHD   |   195.00   | 158.33 |  133.33   | 173.33 |      53.7      |      68.3       |   86.9   |
>
>   - **Takeaways for  prerequisites:**
>     - **Robust to dataset scale:** Performance remains stable across 25%-100% dataset sizes
>     - **Diversity-preferred but robust:** While diverse datasets (PHD) achieve optimal results, even homogeneous datasets (POPE) still improve over baselines.
>
> > ### **About W3: Hyperparameter sensitivity.**
>
> We thank the reviewer for this observation. We would like to clarify the purpose and findings of Figure 3:
> - **Purpose of wide parameter range:** In Figure 3, we intentionally explored an extensive parameter range (including extreme values like α/β up to 10 and negative values) to comprehensively demonstrate the behavioral boundaries of our method. This wide range analysis is meant to provide thorough understanding rather than suggest these extremes are practical choices.
> - **Stable performance in reasonable ranges:** Importantly, our method maintains stable and consistently superior performance compared to baselines across broad reasonable parameter ranges:
>   - α ∈ [0.5, 8] and β ∈ [0.5, 1]: All configurations outperform the baseline and achieve results within 5% of optimal performance
>   - K ∈ [32, 256]: Performance remains stable and consistently better than baselines
>   - Clusters ∈ [2, 5]: All settings yield strong improvements
>
> Sharp performance drops only occur at unreasonable extremes (e.g., α/β > 10 or negative values) that would naturally be excluded during standard tuning procedures.

---

> > ### Author Response · Authors · 2025-11-25
> > **Response to Reviewer KcVa: Part 3**
> >
> > > ### **About Q1: Counterfactual generation**
> >
> > We thank the reviewer for seeking this clarification. We would like to clarify an important distinction:
> > Steering vector construction uses discriminative and multiple-choice  datasets only: Our truthfulness steering vector database is constructed using AMBER(discriminative subset) and SEED datasets, which are discriminative/multiple-choice tasks with clear ground truth answers. For these datasets, counterfactual generation is straightforward and deterministic:
> > - For discriminative questions (yes/no): We flip the answer (yes→no, no→yes)
> > - For multiple-choice questions: We randomly select an incorrect option
> >
> > Open-ended datasets are used only for evaluation: CHAIR and the generative tasks in AMBER are used solely for testing the generalization ability of our method, not for constructing steering vectors. This design choice ensures:
> > - Clean and reliable steering vectors from well-defined positive/negative pairs
> > - Robust evaluation on challenging open-ended generation tasks
> >
> > > ### **About Q2: Automatic hyperparameter selection:**
> >
> > Training-free methods commonly require hyperparameters, such as ICT [12], VTI [13], and DAMO [13]. As discussed, our method shows consistent improvements across a reasonable range of hyperparameter values. In future work, we will explore automatic hyperparameter selection methods, such as
> > - Meta-heuristic optimization: Evolutionary algorithms to efficiently search hyperparameter space
> > - Meta-learning: Training a lightweight predictor that maps model characteristics (architecture, layer depth) to optimal hyperparameters, enabling generalization to new domains.
> >
> > ___
> > **Reference:**
> >
> > [1] Eyes wide shut? exploring the visual shortcomings of multimodal llms. CVPR 2024.
> >
> > [2] MM-Vet: evaluating large multimodal models for integrated capabilities. ICML2024.
> >
> > [3] Illusionvqa: A challenging optical illusion dataset for vision language models. arXiv 2024.
> >
> > [4] Hallusionbench: an advanced diagnostic suite for entangled language hallucination and visual illusion in large vision-language models. CVPR 2024.
> >
> > [5] Qwen2. 5-vl technical report. arXiv 2025.
> >
> > [6] Internvl3: Exploring advanced training and test-time recipes for open-source multimodal models. arXiv 2025.
> >
> > [7] Mantis: Interleaved multi-image instruction tuning. TMLR 2024.
> >
> > [8] Building and better understanding vision-language models: insights and future directions.  arXiv 2024.
> >
> > [9] Evaluating Object Hallucination in Large Vision-Language Models. EMNLP 2023.
> >
> > [10] PhD: A ChatGPT-Prompted Visual Hallucination Evaluation Dataset. CVPR 2025.

---

> ### Author Response · Authors · 2025-11-25
>
> Dear Reviewer KcVa:
>
> We sincerely apologize for the confusion. Due to the sequential submission process of the rebuttal system, our response to your review was submitted separately and may have appeared later. We have prepared a detailed and carefully structured response to address all your questions, which required additional time to organize comprehensively. We greatly value your feedback and appreciate your timely follow-up, which demonstrates your genuine engagement with our work. Hope these explanations and responses could address your concerns. Please let us know if you have any concerns, and we will be more than happy to answer them.

---

### Official Review · Reviewer_uqR8 · 2025-10-23

**Soundness:** 3
**Presentation:** 3
**Contribution:** 3
**Rating:** 4
**Confidence:** 4

**Summary:**

This paper proposes Dynamic Multimodal Activation Steering (DMAS), a training-free and plug-and-play method to reduce hallucinations in Large Vision-Language Models (LVLMs). The authors find that truthfulness and visual perception rely on distinct attention heads and that truthfulness patterns vary with semantics. DMAS builds a semantic truthfulness steering vector database and computes visual perception steering vectors, dynamically applying them to the most relevant attention heads during inference.

**Strengths:**

1. This paper conducts an interesting analysis of attention patterns, revealing which attention heads are most sensitive to truthfulness versus visual perception.
2. This paper proposes an interesting Dynamic Multimodal Activation Steering, which incorporates steering vector to mitigate hallucination.
3. The paper demonstrates good writing quality and is easy to read.

**Weaknesses:**

1. Experiments were conducted on a limited set of backbones; it would be better to include experiments on more recent models.
2. Some important hyperparameters are missing from the paper. For example, the temperature, top-p, and top-k settings are not reported.
3. The paper lacks comparisons with recent decoding strategies, such as DECO [1] and DAMO [2]. As far as I know, they also perform well in hallucination mitigation.
    - [1] MLLM can see? Dynamic Correction Decoding for Hallucination Mitigation
    - [2] DAMO: Decoding by Accumulating Activations Momentum for Mitigating Hallucinations in Vision-Language Models
4. The layout of Table 3 and Table 4 could be improved for better readability and presentation.
5. The formatting of captions should be consistent throughout the manuscript. For example, Figure 1, Figure 2, and Table 1 all end with a period, but Figure 3 does not.

**Questions:**

1. How does the method perform on more recent models? For example, Qwen2.5-VL-7B, which was released at the beginning of 2025.
2. While MME and POPE are simple and well-known benchmarks, how does the method perform on other, more complex ones, such as MM-Vet and LLaVA-Bench? The effectiveness and improvements of the proposed method should be demonstrated across a broader range of benchmarks.
3. Could the authors release the detailed hyperparameter settings?
4. Did the authors use greedy search?
    - If YES, I am a bit curious about the reported results.  Based on my own experiments with LLaVA-1.5-7B on the MME benchmark, the `Existence` subtask under the regular setting should achieve a score around 190, while `Count` and `Color` should be around 160 and 165, respectively.  (Note: Here, I refer to directly using the original MME benchmark rather than data from other repositories, as those may introduce different prompts.)
    - If NOT, multiple runs should be conducted to obtain statistical results and reduce randomness or variance in the evaluation.
5. Based on the reported results, the improvements appear to be marginal on POPE on LLaVA, with performance comparable to ICT. So how to demonstrate the effectiveness?
6. It is interesting that the paper identifies which attention heads are most sensitive to truthfulness versus visual perception. I am curious whether these findings also hold for different models — for example, Qwen-VL, which was used in the main experiments. Furthermore, for the 13B model reported in Table 8, what are the corresponding findings? Since the 7B and 13B models have different numbers of layers, it would be insightful to know whether similar attention patterns are observed.

---

> ### Author Response · Authors · 2025-11-25
> **Response to Reviewer uqR8: Part 1**
>
> We sincerely thank the reviewer for their constructive comments and insightful suggestions, which have been invaluable in improving our manuscript. We are also grateful for your recognition of the strengths of our work. The revisions have been incorporated in the revised manuscript marked in blue. Please kindly find point-to-point responses below.
>
> > ### **About W1 and Q1: Experiments on more backbones.**
>
> Thank you for your valuable suggestion. Following your recommendation, we have conducted additional experiments on several recent models, including Qwen2.5-VL [1], InternVL3 [2], Mantis [3], and Idefics3 [4], evaluating them on the MME and CHAIR benchmarks. The experimental results are presented in the table below, demonstrating that our method remains effective in mitigating hallucinations across these newer models. We have also included these results in the Appendix (A.11) of the revised manuscript.
>
>
> |            |         | MME                 | CHAIR             |                   |
> |:----------:|:-------:|:-------------------:|:-----------------:|:-----------------:|
> | **Model**  | **Method** | **Total Scores↑** | **CHAIR_S↓**     | **CHAIR_I↓**     |
> | Qwen2.5-VL | Vanilla | 698.33              | 47.4              | 10.0              |
> | Qwen2.5-VL | Ours    | **718.33**          | **43.2**          | **9.5**           |
> | InternVL3  | Vanilla | 698.33              | 28.4              | 7.6               |
> | InternVL3  | Ours    | **713.33**          | **26.4**          | **6.9**           |
> | Mantis     | Vanilla | 653.33              | 44.6              | 12.7              |
> | Mantis     | Ours    | **668.33**          | **34.8**          | **9.7**           |
> | Idefics3   | Vanilla | 665.00              | 49.9              | 8.8               |
> | Idefics3   | Ours    | **678.33**          | **45.2**          | **8.2**           |
>
> > ### **About W2 and Q3: Experiments setting.**
>
> In our main experiments, we set temperature=0, top_p=1, and did not use top-k sampling. We have added these details to the revised manuscript (Line 315).
>
> > ### **About W3: Comparison with Deco and DAMO.**
>
> We sincerely thank the reviewer for this valuable suggestion. We have added comparisons with DECO and DAMO on the MME and CHAIR benchmarks. As shown in the table below, our method outperforms both DECO and DAMO on MME. On CHAIR, while our $CHAIR_I$ score is comparable to DECO, our method achieves significantly better performance on $CHAIR_S$. We have incorporated comparisons into Table 1 and Table 3 in the revised manuscript.
>
> |      |           |        | MME      |        |        | CHAIR   |         |
> |:----:|:---------:|:------:|:--------:|:------:|:------:|:-------:|:-------:|
> |      | Existence | Count  | Position | Color  | Total  | CHAIR_S |CHAIR_I |
> | DECO | 185.00       | 153.33 | 118.33   | 155    | 611.66 | 37.8    | **11.1**    |
> | DAMO | 195.00       | 150.00    | **143.33** | 165.00    | 653.33 | 48.8    | 13.4    |
> | Ours | **195.00**   | **158.33** | 133.33 | **173.33** | **659.99** | **30.8** | 11.4 |
>
> > ### **About W4 and W5: Layout and formatting.**
>
> Thank you for pointing this out. We have improved the layout of Table 3 and Table 4, and ensured consistent formatting of all captions in the revised manuscript.
>
> > ### **About Q2: Experiments on more benchmarks.**
>
> Thank you for this valuable suggestion. We have conducted additional experiments on more diverse and complex benchmarks including MM-Vet [5], MMVP [6], IllusionVQA [7], and HallusionBench (multimodal subset) [8] on both LLaVA v1.5 and Qwen-VL. The results are shown in Table below. Our method demonstrates strong performance across these diverse and complex benchmarks. These additional experiments have been included in the Appendix (A.8 Table 9) of the revised manuscript.
>
> |        | MMVet | MMVP  | IllusionVQA  | HallusionBench |
> |:------:|:-----:|:-----:|:------------:|:--------------:|
> | llava  | 38.90 | 30.33 | 32.64        | 50.16          |
> | VTI    | 29.90 | 30.00 | 30.57        | 46.48          |
> | ICT    | 33.40 | 31.00 | 33.10        | 52.67          |
> | Ours   | **39.3** | **54.33** | **35.17** | **53.73**   |
> | QwenVL | 32.9  | 41.00 | 31.72        | 42.38          |
> | Ours   | **34.40** | **43.33** | **33.56** | **42.90**   |

---

> ### Author Response · Authors · 2025-11-25
> **Response to Reviewer uqR8: Part 2**
>
> > ###  **About Q4:**
>
> Yes, we used greedy search in our experiments. Our experimental setup and baseline results align with ICT [9], which also employs greedy search and  'Regular' denotes the original model performance without any decoding strategies.
>
> To address your concern, we have conducted additional experiments under different decoding settings, including Beam Search (beam=2), Nucleus Sampling (temperature=0.5, top_p=0.7) and Greedy Search. The results on MME, MMVP, and CHAIR benchmarks are presented in the table below. Our method consistently demonstrates effective hallucination mitigation across all decoding settings, confirming the robustness of our approach.  We have also included these results in the Appendix (A.10 Table 12) of the revised manuscript.
>
> |                      |         |            |          | MME      |        |              | MMVP        |
> |:--------------------:|:-------:|:----------:|:--------:|:--------:|:------:|:------------:|:-----------:|
> | **Decoding Strategies** | **Method** | **Existence↑** | **Count↑** | **Position↑** | **Color↑** | **Total Scores↑** | **Accuracy↑** |
> | Beam                 | Vanilla | 190.00     | 153.33   | 113.33   | 148.33 | 604.99       | 29.7        |
> | Beam                 | Ours    | **191.67** | **156.66** | **135.00** | **171.66** | **654.99** | **55.1**    |
> | Nucleus              | Vanilla | 190.00     | 147.78   | 123.89   | 150.55 | 612.22       | 32.3        |
> | Nucleus              | Ours    | **190.00** | **150.00** | **130.00** | **159.44** | **629.44** | **53.1**    |
> | Greedy               | Vanilla | 190.00     | 153.33   | 113.33   | 148.33 | 604.99       | 30.3        |
> | Greedy               | Ours    | **195.00** | **158.33** | **133.33** | **173.33** | **659.99** | **54.3**    |
>
> > ###  **About Q5: Effectiveness of our method.**
> Thank you for this observation. POPE primarily evaluates object-level hallucinations with relatively simple question types. Although ICT is specifically designed for object-level hallucinations, our method still outperforms both ICT and VTI in most cases. To further demonstrate our method's effectiveness, we conducted additional experiments on more diverse benchmarks including MMVP, IllusionVQA, HallusionBench, ScienceQA, and ViQuAE. As shown in the table below, our method significantly outperforms ICT across these benchmarks, confirming the effectiveness of our approach.
>
> | Method   | MMVP  | IllusionVQA | HallusionBench | ScienceQA | ViQuAE |
> |:--------:|:-----:|:-----------:|:--------------:|:---------:|:------:|
> | LLaVA1.5 | 30.33 |    32.64    |      50.16     |   52.75   | 43.38  |
> | VTI      | 30.00  |    30.57    |      46.48     |   51.46   | 42.29  |
> | ICT      | 31.00  |    33.10     |      52.67     |   52.95   | 42.47  |
> | Ours     | 54.33 |    35.17    |      53.73     |   62.27   | 56.00   |
>
> > ###  **About Q6: Attention head in different models.**
>
> Thank you for this insightful question. We have conducted detailed analysis across all models used in our experiments. For LLaVA v1.5 7B (32 layers), the most sensitive truthfulness head is located at Layer 30, Head 12, with sensitive heads primarily concentrated in layers 30, 31, 19, and 14. The most sensitive visual perception head is at Layer 31, Head 11, with concentration in layers 21, 31, 26, and 24. Qwen-VL (32 layers) exhibits similar patterns: the most truthfulness-sensitive head is at Layer 31, Head 11 (concentrated in layers 31, 28), while the most visual perception-sensitive head is at Layer 31, Head 24 (concentrated in layers 31, 30). For LLaVA v1.5 13B (40 layers), the patterns scale proportionally with model depth: the most truthfulness-sensitive head is at Layer 39, Head 17 (concentrated in layers 39, 37), and the most visual perception-sensitive head is at Layer 39, Head 37 (concentrated in layers 39, 38, 36). These results demonstrate that the phenomenon that truthfulness and visual perception engage distinct attention head subsets holds across different architectures (LLaVA vs. Qwen-VL) and scales (7B vs. 13B).
>
> ___
> **Reference:**
>
> [1] Qwen2. 5-vl technical report. arXiv 2025.
>
> [2] Internvl3: Exploring advanced training and test-time recipes for open-source multimodal models. arXiv 2025.
>
> [3] Mantis: Interleaved multi-image instruction tuning. TMLR 2024.
>
> [4] Building and better understanding vision-language models: insights and future directions. arXiv 2024.
>
> [5] MM-Vet: evaluating large multimodal models for integrated capabilities. ICML2024.
>
> [6] Eyes wide shut? exploring the visual shortcomings of multimodal llms. CVPR 2024.
>
> [7] Illusionvqa: A challenging optical illusion dataset for vision language models. arXiv 2024.
>
> [8] Hallusionbench: an advanced diagnostic suite for entangled language hallucination and visual illusion in large vision-language models. CVPR 2024.
>
> [9] ICT: Image-Object Cross-Level Trusted Intervention for Mitigating Object Hallucination in Large Vision-Language Models. CVPR 2025.

---

> ### Comment · Reviewer_uqR8 · 2025-11-25
>
> I appreciate the authors’ detailed response. I believe the authors have addressed all of my concerns, and the new experimental results further demonstrate the effectiveness of the proposed method. Initially, the method appeared to be on par with ICT, but the additional experiments clearly highlight its advantages. I recommend including these new results in the revised version of the paper. And all the typos I found before have been revised.
>
> I decide to increase my score.

---

### Official Review · Reviewer_o2L8 · 2025-10-30

**Soundness:** 3
**Presentation:** 3
**Contribution:** 3
**Rating:** 6
**Confidence:** 3

**Summary:**

The goal of this work is to mitigate hallucinations in LVLMs by steering internal activations in a way that respects both visual grounding and factual correctness. The first contribution is an analysis showing that truthfulness and visual perception largely rely on different subsets of attention heads, and that truthfulness directions vary with semantic context. To address this, the paper proposes a training-free method that precomputes steering vectors for the visual and textual pathways and then applies them in a context-aware manner at inference. Concretely, the method clusters prompts into four semantic groups, builds a truthfulness vector per cluster from activation differences between factual and hallucinated answers, derives a visual-perception vector from clean versus noise-corrupted images with object prompts, and then steers the most influential heads using these vectors.

The results show that the approach improves on discriminative VQA benchmarks such as POPE and MME subsets, and on open-ended captioning with CHAIR, while also showing signs of transfer beyond the construction datasets (ScienceQA, ViQuAE). The ablations explore the choice of hyperparameters and cluster size and importance of using both visual and truthfulness vectors.

**Strengths:**

- A training-free and context-aware method to curb hallucinations in LVLMs by nudging a small set of attention heads tied to visual grounding and factuality. It helps on both open-ended captioning (e.g., CHAIR) and VQA-style benchmarks (e.g., POPE/MME).
- The setup is easy to follow: datasets and metrics are spelled out, baselines are sensible, and the main knobs (α, β, and the number of intervened heads K) are reported with the ranges they tried.
- Ablations show both components (truthfulness and visual) matter, dynamic retrieval beats a single fixed vector, and you get reasonable guidance for choices like cluster count and K rather than hand-wavy defaults.

**Weaknesses:**

- Although training-free, the method isn’t truly plug-and-play across LVLMs: the influential-head masks and truthfulness/visual steering vectors must be recomputed for each new model (with α/β/K re-tuned), so deploying on a different backbone requires non-trivial one-time setup rather than drop-in reuse.
- It’s encouraging to see gains without regressions and signs of cross-domain transfer (Table 5). To make the generality claim more convincing, reporting results against stronger baselines beyond Regular would make the narrative stronger.

**Questions:**

- Given that the steering database is built from AMBER/SEED (discriminative and MCQ) [lines 172-173] using label flips or random incorrect choices, clustered into four groups, I am curious to know the intuition behind how these vectors remain valid for open-ended generation. What mechanism supports transfer from discriminative supervision to generative benchmarks such as CHAIR?
- For visual steering vector, does the selecting strategy for negative objects impact this method’s performance? e.g would selection of hard negatives or commonly hallucinated objects as negatives instead of random negative object selection done in the paper impact performance? What are the takeaways from practitioners on the sensitivity of negative object/answer selection for the database generation?

---

> ### Author Response · Authors · 2025-11-25
> **Response to Reviewer o2L8**
>
> We sincerely thank the reviewer for their constructive comments and insightful suggestions, which have been invaluable in improving our manuscript. We are also grateful for your recognition of the strengths of our work. The revisions have been incorporated in the revised manuscript marked in blue. Please kindly find point-to-point responses below.
>
> > ### **About W1:**
>
> We thank the reviewer for this valuable suggestion. We have revised the expression of "plug-and-play" in the manuscript (Line 18\~19, 58\~60, 131\~132, 480\~481).
>
> > ### **About W2:**
>
> Thank you very much for your suggestion. We have added the performance of VTI and ICT on ScienceQA and ViQuAE. The experimental results are shown in the table below, demonstrating that our model outperforms both VTI and ICT on these two datasets. The corresponding experimental results have also been included in Table 5 of the revised manuscript.
>
> | Method   | ScienceQA | ViQuAE |
> |:--------:|:---------:|:------:|
> | LLaVA1.5 |   52.75   | 43.38  |
> | VTI      |   51.46   | 42.29  |
> | ICT      |   52.95   | 42.47  |
> | Ours     |   62.27   | 56.0   |
>
> > ### **About Q1:**
>
> The Steering Vector is computed by  the non-hallucinated activations minus the hallucinated activations. This direction of the steering vector is from hallucination toward non-hallucination. The essence of using the Steering Vector is to push the internal activations of LVLMs from the hallucinated direction toward the non-hallucinated direction. When dealing with different tasks, the approach consistently guides the model toward producing fewer hallucinations.
>
> > ### **About Q2:**
>
> We thank the reviewer for this insightful question. We compared two selection strategies: (1) random selection across categories (e.g., people → panda), and (2) hard negatives within the same category (e.g., runner → swimmer). We clarify that "random selection" in our paper means random sampling within the same object category (hard negatives), which we have now made explicit in the revised manuscript (line 222\~223).
> As shown in the table below, experiments on MME and CHAIR benchmarks demonstrate that both strategies achieve comparable performance with no significant difference. This indicates that our visual steering vector approach is robust to the choice of negative selection strategy.  This experiment has been incorporated into the Appendix (A.12) of the revised manuscript.
>
> **Takeaways:** The method does not exhibit strong sensitivity to negative selection strategies, making it accessible and easy to implement without requiring careful curation of category-specific hard negatives.
>
> | Strategy      |          MME           |        |          |        |          |      CHAIR      |          |
> |:-------------:|:----------------------:|:------:|:--------:|:------:|:--------:|:---------------:|:--------:|
> |               |     Existence↑     |Count↑|Position↑|Color↑|Total Scores↑|  CHAIR_S↓  |CHAIR_I↓|
> | Random        |         195.00         | 158.33  |  133.33  | 173.33  |  659.99  |       31.2      |   10.9   |
> | Hard Negative |         195.00         | 158.33  |  133.33  | 173.33  |  659.99  |       30.8      |   11.4   |

---

> > ### Comment · Reviewer_o2L8 · 2025-11-28
> >
> > Thank you for your detailed response and incorporating provided suggestions. I will keep my score.

---

### Official Review · Reviewer_kS9v · 2025-11-01

**Soundness:** 2
**Presentation:** 2
**Contribution:** 2
**Rating:** 4
**Confidence:** 4

**Summary:**

This paper introduces Dynamic Multimodal Activation Steering (DMAS), a training-free and plug-and-play method designed to mitigate hallucinations in Large Vision-Language Models (LVLMs). The core idea is to dynamically intervene in attention head activations by separating intervention into two components: a "truthfulness steering vector" and a "visual perception steering vector." The truthfulness vector is learned via contrastive activation differences across semantically clustered data and stored in a dynamic database, while the visual perception vector is derived from clean vs. noisy image inputs. During inference, DMAS retrieves the most context-relevant truthfulness vector and applies both to the top-K most influential attention heads.

**Strengths:**

The paper is well-motivated and proposes a clever approach to VLM hallcunation issue. The realization that truthfulness steering vectors vary significantly across semantic contexts and the resulting dynamic, context-aware database approach is novel within activation steering literature. In addition, the training-free nature of this work is highly preferred, especially for such kind of problems where efficiency matters.

**Weaknesses:**

I am giving a conditional weak reject, and I think the the following issue should be carefully addressed by authors.

- A more comprehensive analysis of the constructed dataset should be provided. One of the major novelty of the method is to pre-construct a set of embeddings and select steering vectors accordingly from this dataset. As such, the property of the dataset matters a lot, but very limited analysis is given. How different the performance will be if we start from a different dataset? Will the size of the dataset matters? How random the performance is when we change the selection criteria? I think the study of dataset is largely lacking, making the understanding of DMAS very restricted.
- The overall improvement shown in the main tables are marginal, and authors are presenting them in a somewhat misleading way (the $\Delta$ should be with the best baselines, not the "Regular". This happens a lot in the paper and must be addressed). In table 2 other methods can give SOTA performance or very close performance. Without standard deviation reported, I am not convinced that this minor improvement over ICV and VTI is significant.  The authors are encouraged to include more experiments to demonstrate the effectiveness of DMAS, or otherwise, I doubt the necessity of having such a dynamic, retrival-related method.

**Questions:**

I think the authors do not use a correct citation format. Citations should be with brackets when the cited papers are not the subjects. please address.

---

> ### Author Response · Authors · 2025-11-25
> **Response to Reviewer kS9v: Part 1**
>
> We sincerely thank the reviewer for their constructive comments and insightful suggestions, which have been invaluable in improving our manuscript. We are also grateful for your recognition of the strengths of our work. The revisions have been incorporated in the revised manuscript marked in blue. Please kindly find point-to-point responses below.
>
> > ### **About W1：Comprehensive analysis of the constructed dataset.**
>
> Thank you very much for your valuable suggestion. We will address your concern in three parts, with the corresponding experiments included in the appendix of the revised manuscript (Appendix A.9, line 864).
> - **How different the performance will be if we start from a different dataset?**
>
>   To explore how different datasets affect steering vector construction, we built databases using two alternative datasets: POPE [1] and PHD [2]. POPE contains only object existence questions (homogeneous), while PHD includes diverse question types similar to our original datasets. Both databases were constructed using the same scale as our main experiments, and we evaluated performance on MME, AMBER, and MMVP [3].
>   Results show that the POPE-based database underperforms on several MME subtasks and AMBER's relation subtask, which is expected given POPE's limited diversity cannot fully exploit our dynamic approach. In contrast, the PHD-based database achieves comparable results to our original method across all benchmarks. These findings demonstrate two important insights: (1) when using datasets with good question diversity, our method achieves consistent performance regardless of the specific dataset chosen, and (2) this validates the necessity of our dynamic retrieval mechanism for selecting appropriate steering vectors based on semantic context.
>
> |       |           MME           |        |          |        |      MMVP      |      AMBER      |          |
> |:-----:|:-----------------------:|:------:|:--------:|:------:|:--------------:|:---------------:|:--------:|
> | Scale | Existence↑ | Count↑ | Position↑ | Color↑ | Accuracy↑ | Relation↑ | F1↑ |
> | Ours  |   195.00   | 158.33 |  133.33   | 173.33 |      54.3      |      69.0       |   87.2   |
> | POPE  |   195.00   | 158.33 |  123.33   | 153.33 |      52.3      |      65.1       |   86.0   |
> | PHD   |   195.00   | 158.33 |  133.33   | 173.33 |      53.7      |      68.3       |   86.9   |
>
>
> - **Will the size of the dataset matters?**
>
>   To investigate the impact of dataset size on constructing steering vectors, we systematically reduced the dataset to 75%, 50%, and 25% of its original size and conducted experiments on three benchmarks: MME, AMBER, and MMVP . The results are presented in the table below.
>   As shown in the results, the dataset size does not significantly affect the performance of our method. The fluctuations across the three benchmarks remain minimal, demonstrating the robustness of our approach with respect to dataset scale.
>
>
> |       |          MME           |        |          |        |      MMVP      |      AMBER      |          |
> |:-----:|:-----------------------:|:------:|:--------:|:------:|:--------------:|:---------------:|:--------:|
> | Scale | Existence↑ | Count↑ | Position↑ | Color↑ | Accuracy↑ | Relation↑ | F1↑ |
> | 100%  |   195.00   | 158.33 |  133.33   | 173.33 |      54.3      |      69.0       |   87.2   |
> | 75%   |   195.00   | 158.33 |  133.33   | 168.33 |      52.0      |      69.0       |   87.2   |
> | 50%   |   195.00   | 158.33 |  133.33   | 168.33 |      52.0      |      69.0       |   87.2   |
> | 25%   |   195.00   | 158.33 |  133.33   | 168.33 |      53.0      |      69.0       |   87.1   |
>
> - **How random the performance is when we change the selection criteria?**
>   Our experiments demonstrate that DMAS exhibits low sensitivity to dataset selection:
>   - **Robust to dataset scale**: Performance remains stable across 25%-100% dataset sizes.
>   - **Diversity-preferred but robust**: While diverse datasets (PHD) achieve optimal results, even homogeneous datasets (POPE) still improve over baselines.

---

> ### Author Response · Authors · 2025-11-25
> **Response to Reviewer kS9v: Part 2**
>
> > ### **About W2: More comparison with ICV and VTI.**
>
> - **Regarding $\Delta$:**
>
>   We appreciate the suggestion. We have removed $\Delta$ from the table to avoid potential confusion in the revised manuscript.
>
> - **Regarding improvement in Table 2:**
>
>   Table 2 presents our results on POPE, a benchmark specifically designed for evaluating object-level hallucinations, which features a relatively homogeneous question format. Although ICT  incorporates designs for object-level hallucination, our method outperforms both ICT and VTI in most cases. To further substantiate the effectiveness of our approach, we conducted extensive experiments on a broader and more diverse set of benchmarks: MMVP [3], IllusionVQA [4], HallusionBench (mulrimodal subset) [5], ScienceQA, and ViQuAE. The results are summarized in the table below, demonstrate that our method achieves consistently superior performance over ICT and VTI across these benchmarks, validating the efficacy of our dynamic, retrieval-related method. These comparisons have been incorporated into Table 5 and the Appendix (A.8) in revised manuscript.
>
> | Method   | MMVP  | IllusionVQA | HallusionBench | ScienceQA | ViQuAE |
> |:--------:|:-----:|:-----------:|:--------------:|:---------:|:------:|
> | LLaVA1.5 | 30.33 |    32.64    |      50.16     |   52.75   | 43.38  |
> | VTI      | 30.00  |    30.57    |      46.48     |   51.46   | 42.29  |
> | ICT      | 31.00  |    33.10     |      52.67     |   52.95   | 42.47  |
> | Ours     | **54.33** |    **35.17**    |      **53.73**     |   **62.27**   | **56.00**   |
>
> > ### **About Q1: Citation formatting.**
>
> Thanks for your careful review. We have thoroughly addressed and corrected the citation formatting throughout the revised manuscript.
> ___
> **Reference:**
>
> [1] Evaluating Object Hallucination in Large Vision-Language Models. EMNLP 2023.
>
> [2] PhD: A ChatGPT-Prompted Visual Hallucination Evaluation Dataset. CVPR 2025.
>
> [3] Eyes wide shut? exploring the visual shortcomings of multimodal llms. CVPR 2024.
>
> [4] Illusionvqa: A challenging optical illusion dataset for vision language models. arXiv 2024.
>
> [5] Hallusionbench: an advanced diagnostic suite for entangled language hallucination and visual illusion in large vision-language models. CVPR 2024.

---

### Author Response · Authors · 2025-12-02
**Summary of Discussion: Part 1**

We regret that we cannot engage in further discussion with all reviewers. We sincerely thank the AC and PC for their dedication to our paper. We are grateful for reviewers' recognition of the strengths of our work and the insightful suggestions. During the rebuttal period, we provided detailed responses to each reviewer's concerns and made corresponding revisions to the paper, which are highlighted in blue. **Three out of four reviewers participated in the discussion**, and the **scores improved from (6/6/4/4) to (6/6/6/4)**.
- **Reviewer uqR8 stated, 'I believe the authors have addressed all of my concerns' and raised score from 4 to 6 in Nov 26.**
- **Reviewer KcVa's final score is 6 and increased confidence from 3 to 4 in Nov 26.**
- **Reviewer o2L8 kept  positive score of 6.**
- While we regret that Reviewer kS9v could not participate in the discussion, we have provided comprehensive responses to each concern and question raised and we have successfully addressed all the issues.

We have prepared this summary to provide a concise overview of the review process and the significant revisions made during the rebuttal period.

### **Summary of Strengths**

Here is a summary of strengths of our papers mentioned by reviewers:
- **The paper is well-motivated and conducts an interesting analysis of attention patterns in LVLMs.**
  - The paper is well-motivated and proposes a clever approach to VLM hallcunation issue. (Reviewer kS9v)
  - The paper provides an in-depth analysis of hallucination issues in LVLMs, particularly revealing the functional tendencies of different attention heads and the semantic dependence of steering vectors through preliminary studies, which offers strong motivation for the subsequent method design. (Reviewer KcVa)
  - This paper conducts an interesting analysis of attention patterns, revealing which attention heads are most sensitive to truthfulness versus visual perception. (Reviewer uqR8)
- **The proposed method DMAS is novel and interesting.**
  - The realization that truthfulness steering vectors vary significantly across semantic contexts and the resulting dynamic, context-aware database approach is novel within activation steering literature. （Reviewer kS9v）
  - This paper proposes an interesting Dynamic Multimodal Activation Steering, which incorporates steering vector to mitigate hallucination. （Reviewer uqR8）
- **Training-free nature of this work is highly preferred.**
  - In addition, the training-free nature of this work is highly preferred, especially for such kind of problems where efficiency matters. (Reviewer kS9v)
  - A training-free and context-aware method to curb hallucinations in LVLMs by nudging a small set of attention heads tied to visual grounding and factuality. (Reviewer o2L8)
- **Broad experimental coverage and significant performance improvement.**
  - The paper conducts comprehensive experiments across multiple models (LLaVA v1.5, QwenVL) and multiple benchmarks (MME, POPE, CHAIR), including detailed ablation studies and case analyses, demonstrating the effectiveness of the method. (Reviewer KcVa)
  - Experimental results show that DMAS achieves state-of-the-art performance across multiple tasks, with particularly substantial gains on MME and CHAIR metrics. (Reviewer KcVa)
  - It helps on both open-ended captioning (e.g., CHAIR) and VQA-style benchmarks (e.g., POPE/MME). (Reviewer o2L8)
  - Ablations show both components (truthfulness and visual) matter, dynamic retrieval beats a single fixed vector, and you get reasonable guidance for choices like cluster count and K rather than hand-wavy defaults. (Reviewer o2L8)
- **The paper demonstrates good writing quality and setup is easy to follow.**
  - The paper demonstrates good writing quality and is easy to read. (Reviewer uqR8)
  - The setup is easy to follow: datasets and metrics are spelled out, baselines are sensible, and the main knobs (α, β, and the number of intervened heads K) are reported with the ranges they tried. (Reviewer o2L8)

### **Summary of Rebuttal**
### **Reviewer uqR8：**
- **Concerns and How We Resolved**：
  - Experiments on more backbones.
    - We validated that DMAS still achieves improvements on advanced models including Qwen2.5-VL, InternVL3, Mantis, and Idefics3, as demonstrated by evaluations on MME and CHAIR benchmarks (Appendix A.11).
  - About experiment setting.
    - We clarified our experimental setup in the revised manuscript (line 315), and provided additional experiments demonstrating that DMAS achieves strong performance across different decoding strategies (Appendix A.10).
  - Comparison with Deco and DAMO.
    - We added these two baselines and verified that our method outperforms both on MME and CHAIR benchmarks.
  - Experiments on more benchmarks.
    - We conducted additional experiments on MM-Vet, MMVP, IllusionVQA, and HallusionBench (multimodal subset) using both LLaVA v1.5 and Qwen-VL, validating the effectiveness of DMAS (Appendix A.8).

---

> ### Author Response · Authors · 2025-12-02
> **Summary of Discussion: Part 2**
>
> - **Concerns and How We Resolved:**
>   - Effectiveness of our method.
>     - We compared DMAS with ICT and VTI on MMVP, IllusionVQA, HallusionBench, ScienceQA, and ViQuAE to validate our model's effectiveness. DMAS significantly outperforms ICT and VTI on these more diverse and challenging benchmarks. (Table 5 and Appendix A.8).
> - **Results**: Reviewer uqR8 stated **'I believe the authors have addressed all of my concerns'** and **raised score from 4 to 6 in Nov 26**.
>
> ### **Reviewer KcVa:**
> - **Concerns and How We Resolved：**
>   - Bias in constructed steering vectors.
>     - Although our steering vector is built on discriminative and multiple-choice questions, Table 3 shows its strong effect on open-ended generation, and Table 5 validates it on subject/knowledge-based QA. This demonstrates its generalization. This transferability can be attributed to the underlying mechanism of our approach. The Steering Vector represents the direction from hallucinated to non-hallucinated activations in the model's representation space. By steering activations along this direction, we guide the model toward producing more faithful outputs regardless of task format.
>   - About Generality, Scalability, and Prerequisites.
>     - For generality, Table 5 in the original paper has demonstrated the generality of DMAS. We further supplemented results on more benchmarks to further prove this. The experimental results show that DMAS achieves strong performance across these diverse benchmarks (Appendix A.8).
>     - For scalability, Table 8 in the appendix demonstrates its scalability across model sizes, and our supplementary experiments further confirm its scalability across different model architectures (Appendix A.11).
>     - For prerequisites, we investigated the impact of data scale and dataset style. Our findings show that DMAS is robust to both factors, while its advantages are most fully realized with datasets of high diversity (Appendix A.9).
>   - Hyperparameter sensitivity and automatic hyperparameter selection
>     - We clarified that Figure 3 intentionally explores an extensive parameter range to demonstrate our method's behavioral boundaries. Within reasonable ranges, our method consistently outperforms the baseline. We also listed automatic parameter selection methods.
> - **Results**: A misunderstanding initially occurred with the reviewer,  due to the sequential nature of the rebuttal submission process. This was constructively resolved through our rebuttal, as evidenced by the reviewer’s final assessment on Nov 26. **Reviewer KcVa's final score is 6 and increased confidence from 3 to 4 on Nov 26.**
>
> ### **Reviewer o2L8：**
> - **Concerns and How We Resolved：**
>   - About expressions on 'plug and play'.
>     -  We have revised the expression of "plug-and-play" in the manuscript (Line 18\~19, 58\~60, 131\~132, 480\~481).
>   - Comparision with ICT and VTI on ScienceQA and ViQuAE.
>     - We have added the performance of VTI and ICT on ScienceQA and ViQuAE. The experimental results demonstrate that our model outperforms both VTI and ICT on these two datasets (Table 5).
>   - About negative objects selection strategis.
>     - We compared random selection and Hard Negative. The experimental results show that the performance under these two settings is comparable, indicating that our method is robust to the choice of negative selection strategy (Appendix A.12).
> - **Results**: Reviewer o2L8 **kept positive score of 6.**
>
> ### **Reviewer kS9v：**
> - **Concerns and How We Resolved：**
>   - Comprehensive analysis of the constructed dataset.
>     - We have conducted two additional sets of experiments to investigate the impact of data scale and dataset style on DMAS, respectively. The results yield two key findings. First, DMAS is robust to dataset scale, as its performance remains stable when using 25% to 100% of the training data. Second, while DMAS achieves optimal results on diverse datasets (e.g., PHD), it still yields significant improvements over baselines even on homogeneous datasets (e.g., POPE). These findings collectively demonstrate the robustness of DMAS. Moreover, the superior performance on diverse data underscores the necessity of dynamically invoking different semantic vectors (Appendix A.9).
>   - More comparison with ICV and VTI.
>     - We compared our method with ICT and VTI on a broader and more diverse set of benchmarks: MMVP, IllusionVQA, HallusionBench, ScienceQA, and ViQuAE. The results demonstrate that our method achieves consistently superior performance over ICT and VTI across all these benchmarks, validating the efficacy of our dynamic, retrieval-related method (Table 5 and Appendix A.8).
>   - Citation formatting.
>     - We have thoroughly addressed and corrected the citation formatting throughout the revised manuscript.
> - **Results**: No reply before review roll-back.

---

### Meta-Review · Area_Chair_wrFd · 2025-12-03

**Summary:**

This paper proposes Dynamic Multimodal Activation Steering (DMAS), a training-free method to reduce hallucinations in Large Vision-Language Models. The approach is based on two key observations: different attention heads handle truthfulness and visual perception, and truthfulness steering vectors are semantically dependent. DMAS dynamically applies pre-computed steering vectors to influential attention heads during inference. Initial reviewer scores were 6, 6, 4, and 4. The main concerns included the need for more comprehensive analysis of the steering vector dataset, comparisons with recent baselines (e.g., DECO, DAMO), demonstrations of effectiveness and generality across more models and benchmarks, and discussions on hyperparameter sensitivity and prerequisites. In the rebuttal, the authors provided extensive additional experiments. They validated DMAS on newer models (Qwen2.5-VL, InternVL3), added comparisons with DECO and DAMO, and showed strong results on more diverse benchmarks (MM-Vet, MMVP). They also conducted ablation studies on dataset scale, diversity, and negative sample selection, demonstrating robustness. The rebuttal successfully addressed most concerns, leading to score improvements from reviewers uqR8 (4->6) and KcVa (6 with increased confidence), while reviewer o2L8 maintained a score of 6. Reviewer kS9v did not participate in the discussion, but the authors provided detailed responses to their points.

**Reviewer Concerns:**

The authors have adequately addressed the major concerns regarding dataset analysis, additional comparisons, and generality through new experiments. The remaining minor point is that practical deployment still requires a one-time, per-model setup for vector computation and hyperparameter tuning.

**Reviewer Scores:**

Reviewer uqR8 explicitly raised their score from 4 to 6 after being satisfied with the new experiments on more models and benchmarks. Reviewer KcVa increased their confidence level after their concerns about generality and prerequisites were addressed through the new scalability and robustness analyses. Reviewer o2L8 maintained their positive score of 6. While reviewer kS9v did not participate in the discussion, the comprehensive responses aimed at addressing their concerns likely would have led to a score increase.

---

### Decision · Program_Chairs · 2026-01-26

Accept (Poster)